# Bridging multiscale interfaces for developing ionically conductive high-voltage iron sulfate-containing sodium-based battery positive electrodes

Jiyu Zhang [1,4], Yongliang Yan[1,4], Xin Wang[1], Yanyan Cui[2], Zhengfeng Zhang [3], Sen Wang[1], Zhengkun Xie[1], Pengfei Yan [3] & Weihua Chen [1] ✉

Non-aqueous sodium-ion batteries (SiBs) are a viable electrochemical energy storage system for grid storage. However, the practical development of SiBs is hindered mainly by the sluggish kinetics and interfacial instability of positive-electrode active materials, such as polyanion-type iron-based sulfates, at high voltage. Here, to circumvent these issues, we proposed the multiscale interface engineering of $Na_{2.26}Fe_{1.87}(SO_4)_3$, where bulk heterostructure and exposed crystal plane were tuned to improve the Na-ion storage performance. Physicochemical characterizations and theoretical calculations suggested that the heterostructure of $Na_6Fe(SO_4)_4$ phase facilitated ionic kinetics by densifying Na-ion migration channels and lowering energy barriers. The (11-2) plane of $Na_{2.26}Fe_{1.87}(SO_4)_3$ promoted the adsorption of the electrolyte solution $ClO_4^-$ anions and fluoroethylene carbonate molecules, which formed an inorganic-rich Na-ion conductive interphase at the positive electrode. When tested in combination with a presodiated FeS/carbon-based negative electrode in laboratory- scale single-layer pouch cell configuration, the $Na_{2.26}Fe_{1.87}(SO_4)_3$-based positive electrode enables an initial discharge capacity of about 83.9 mAh g$^{-1}$, an average cell discharge voltage of 2.35 V and a specific capacity retention of around 97% after 40 cycles at 24 mA g$^{-1}$ and 25 °C.

Environmental pollution and limited fossil energy push the development of renewable clean energies. However, their intermittent and regional features, which are prioritized at the forefront and in projects of countries, require large-scale energy storage systems to function together[1]. Therefore, sodium-ion batteries (SiBs), as one of the most promising next-generation energy storage technology, have introduced a new opportunity due to the abundant Na reserves and their worldwide distribution[2]. The development of high-voltage (>3.6 V vs. Na$^+$/Na) and resource-abundant cathodes with long operating life to

circumvent raw and cost limitations is progressing rapidly[3,4]. In this regard, alluaudite-type iron-based sulfates are considered promising positive-electrode active material candidates due to their abundant resources, high voltage (3.8 V vs. Na$^+$/Na), and robust polyanion frameworks[5]. However, the large bandgap hinders the rapid transfer and reaction kinetics of charge carriers (Na$^+$ and electron), which results in the poor cycling reversibility and rate performance of batteries[6]. Furthermore, electrolytes suffer from severe oxidative decomposition at high voltages, which causes the formation of a solid

[1]College of Chemistry & Green Catalysis Center, Zhengzhou University, Zhengzhou 450001 Henan, China. [2]Institute of Nanotechnology, Karlsruhe Institute of Technology (KIT), Hermann-von-Helmholtz-Platz 1, 76344 Eggenstein Leopoldshafen, Germany. [3]Beijing Key Laboratory of Microstructure and Properties of Solids, Faculty of Materials and Manufacturing, Beijing University of Technology, Beijing 100124, China. [4]These authors contributed equally: Jiyu Zhang, Yongliang Yan. ✉e-mail: chenweih@zzu.edu.cn

interphase, that is, the cathode electrolyte interphase (CEI)[7,8]. CEI influences Na$^+$ insertion from the solvated phase into the solid phase, and thus constitutes an extra rate-limiting step for sodiation of most cathodes.

Highly connected ion-conducting pathways within the cathode may provide an effective strategy to circumvent the above problems[9–11]. Their construction requires the consideration of all essential components in the positive electrode, including particles of active materials and their compatibility with electrolytes at the cell level, because ion-conducting pathways are distributed at all length scales, from Na$^+$ migration within crystalline lattices or boundaries to Na$^+$ accommodation or dislocation within the CEI at the cathode interface[12]. For the bulk design of particles, nano-engineering of cathode materials is usually used to decrease the ionic transfer distance[13,14]. However, the effect is upper-limited by electrode materials, which cannot accommodate the Na$^+$ rapidly passing through during solid-solution transformation due to large kinetic barriers and limited migration channels available inside the crystals. The introduction of additional materials with high ionic conductivity was used to facilitate the ionic kinetics of composite cathodes, such as the tailored heterostructure in oxide cathodes, through the incorporation of semiconductor components with different bandgaps[15,16]. However, the chemical construction of heterostructures in positive electrodes with complex polyanion frameworks has not been reported and, is subject to harsh synthesis conditions such as relatively low temperature (<350 °C) and uncontrollable solid-phase reaction thermodynamics[17]. In this sense, the heterostructure design of bulk materials still needs to be explored to obtain satisfactory performance on high-voltage cathodes.

The primary construction of CEI is intrinsically dependent on electrochemical reactions of the electrolyte on the cathode surface[18,19]. Typically, constrained by the strong surface nucleophilicity of the cathode, solvents based on ethylene carbonate are preferentially absorbed, and the derived oligomers (e.g., polycarbonate) constitute a thick and ionically insulating CEI through nucleophilic reactions, which increases the interfacial impedance and battery polarization. Thus, the exposed crystal planes of cathode materials, which are closely related to their electronic structure and atomic properties, are one of the key parameters to manipulate the CEI formation[20]. However, the influence on CEI formation of the cathodes, specifically their exposed crystal planes and electric structures, is not fully investigated[21].

In this work, multiscale interfaces were successfully integrated in high-voltage iron-based sulfate cathodes to build continuous Na$^+$ transfer channels at all length scales. Specifically, the ionic kinetics of Na$_{2.26}$Fe$_{1.87}$(SO$_4$)$_3$ inside the particle bulk was improved by introducing minor ionic-conductive Na$_6$Fe(SO$_4$)$_4$ phase with dense Na$^+$ migration channels and low barriers. The rapid ionic transfer across the cathode/electrolyte interface and electrolyte stability at high voltage were achieved via an inorganic-rich and uniform CEI, which was induced by the (11-2) plane exposure of Na$_{2.26}$Fe$_{1.87}$(SO$_4$)$_3$ with a low-electron density. Their synergy qualitatively improved the ionic kinetics and reversibility of the positive-electrode active material, enabling its successful application in lab-scale iron sulfur-based Na-ion cells to demonstrate the scalability of these class of materials to up to a technology readiness level of 4[22].

## Results and discussion

A sodium-ion-conducting high-voltage cathode Na$_{2.26}$Fe$_{1.87}$(SO$_4$)$_3$ phase with a Na$_6$Fe(SO$_4$)$_4$ phase heterostructure and an electronically conductive carbon network (named NFS-H) was synthesized via co-precipitation and subsequent calcination (Supplementary Fig. 1). Na$_2$SO$_4$ and FeSO$_4$·7H$_2$O were dissolved and coprecipitated into hydrated sulfate (Na$_2$Fe(SO$_4$)$_2$·4H$_2$O) as intermediate during the alcohol-induced precipitation and another intermediate thermally decomposed into off-stoichiometric Na$_{2+2x}$Fe$_{2-x}$(SO$_4$)$_3$ with a

heterostructure of Na$_6$Fe(SO$_4$)$_4$ through thermal competition reactions due to its local atomic inhomogeneity[23]. Therefore, heterostructured phases can be generated synchronously with close contact (Fig. 1a). Importantly, the dominant crystal plane of the intermediate (Na$_2$Fe(SO$_4$)$_2$·4H$_2$O) featured a high similarity to the (11-2) plane of Na$_{2+2x}$Fe$_{2-x}$(SO$_4$)$_3$ in atomic plane arrangement, which eased the transformation of the former into the latter atomic arrangements through topological reactions. (Supplementary Fig. 2)[24,25]. By calculating the various dominant crystal plane of Na$_{2+2x}$Fe$_{2-x}$(SO$_4$)$_3$ crystals, we determined that the (11-2) plane had a relatively loose atomic arrangement and the lowest electron density with surface energy (Supplementary Fig. 2). Thus, the obtained Na$_{2+2x}$Fe$_{2-x}$(SO$_4$)$_3$ may be more likely to expose the (11-2) crystal plane, and the products directly converted from Na$_2$SO$_4$ and FeSO$_4$ exhibited weaker diffraction intensities[26,27]. NFS-H particles featured micron-sized (6 – 8 μm) polyhedra inherited from the intermediate, and they consisted of stacked nanoparticles and multilayer reduced graphene oxide sheets attached to the surface (the inset in Fig. 1b and Supplementary Fig. 3). The micron-sized particles with a low surface area (8.15 m$^2$ g$^{-1}$) and meso- and microporous structures allowed structural robustness and sufficient electrolyte penetration (Supplementary Fig. 4)[28].

All the diffraction peaks in the synchrotron high-pressure powder X-ray diffraction (XRD) pattern of NFS-H (Fig. 1b) can be assigned to the alluaudite-type Na$_{2.26}$Fe$_{1.87}$(SO$_4$)$_3$ and vanthoffite-type Na$_6$Fe(SO$_4$)$_4$, in which the sharp diffraction peaks indicated a high crystallinity. Notably, compared with Na$_{2+2x}$Fe$_{2-x}$(SO$_4$)$_3$ which was converted directly from the composite of Na$_2$SO$_4$ and FeSO$_4$ as reported in literature[26,27], the intermediate-derived Na$_{2.26}$Fe$_{1.87}$(SO$_4$)$_3$ showed a relatively stronger (11-2) peak but weaker (200) peaks. Moreover, the contents of Na$_{2.26}$Fe$_{1.87}$(SO$_4$)$_3$ and Na$_6$Fe(SO$_4$)$_4$ were adjusted to 90.67 and 9.33 wt% by Rietveld refinement, respectively (detailed crystallographic parameters in Supplementary Tables 1–4). Within the Na$_{2.26}$Fe$_{1.87}$(SO$_4$)$_3$ crystal, all the Na$^+$ can be assigned to three types (Na1, Na2, and Na3) based on their individual binding energies, whereas the Fe$_2$O$_{10}$ dimer is composed of two edge-sharing FeO$_6$ octahedra and SO$_4$ tetrahedra units through corner-sharing oxygen atoms (Supplementary Fig. 5). Meanwhile, for the Na$_6$Fe(SO$_4$)$_4$ crystal, the host lattice consisted of FeO$_6$ octahedra and SO$_4$ tetrahedra through corner-sharing oxygen atoms, in the voids containing three types of Na sites. The similar physicochemical properties of elements and crystal unit parts in the two phases, especially the identical polyhedrons in their frameworks, offer a great possibility for heterostructures during synchronous configuration. For comparison, a Na$_{2+2x}$Fe$_{2-x}$(SO$_4$)$_3$ material containing the less ionic-conductive Na$_6$Fe(SO$_4$)$_4$ phase (4.52 wt%) was synthesized (named as NFS-L, Supplementary Fig. 6) by promoting the single-phase precipitated FeSO$_4$·4H$_2$O to regulate the Na$_2$SO$_4$-FeSO$_4$ reaction balance[29]. However, given the uncontrollable thermodynamics and metastable properties, pure Na$_{2.26}$Fe$_{1.87}$(SO$_4$)$_3$ and Na$_6$Fe(SO$_4$)$_4$ materials were difficult to obtain from the intermediate.

To gain insights into the intrinsic properties of heterostructured phases in NFS-H in ionic transfer kinetics, we macroscopically probed Na$^+$ migration pathways by bond valence site energy (BVSE) calculation and visualized them as regions enclosed by isosurfaces. The calculation workflow was managed by the high-throughput computational platform for battery materials[30,31]. Visibly, the Na$_{2.26}$Fe$_{1.87}$(SO$_4$)$_3$ crystal provided intermittent and narrow channels for Na$^+$ migration along the b and c-axis directions across a zigzag path between two equivalent positions (Fig. 1c and Supplementary Fig. 7). The punctate one-dimensional (1D) pathways limited the rapid Na$^+$ migration inside the crystal, which may account for the poor ionic conductivity of Na$_{2+2x}$Fe$_{2-x}$(SO$_4$)$_3$ materials. Differently, the Na$_6$Fe(SO$_4$)$_4$ crystal possessed successive and broad Na$^+$ migration pathways along the 3D (a, b, and c-axis) directions (Fig. 1d). Further calculations of the energy barrier revealed that Na$^+$ migrated mainly along the 1D and 2D

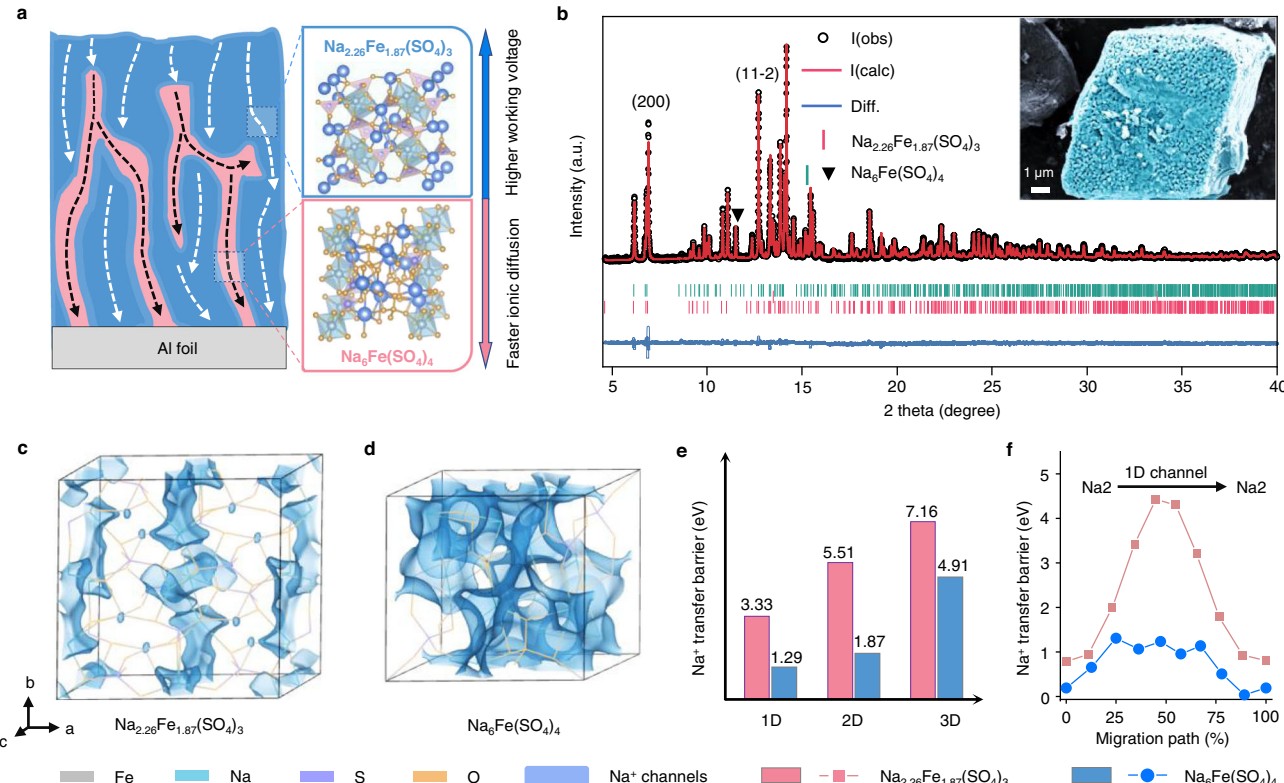

**Fig. 1 | Structural characterizations and Na⁺ transfer investigation of as-synthesized heterostructured NFS-H. a** Schematic illustration of the heterostructure. The blue and red areas represent $Na_{2.26}Fe_{1.87}(SO_4)_3$ and $Na_6Fe(SO_4)_4$ in positive material, respectively. The white and black arrows represent sodium-ion transport in $Na_{2.26}Fe_{1.87}(SO_4)_3$ and $Na_6Fe(SO_4)_4$, respectively. And the blue and red arrows represent the characteristics of $Na_{2.26}Fe_{1.87}(SO_4)_3$ (higher working voltage) and $Na_6Fe(SO_4)_4$ (faster ionic diffusion), respectively. **b** Synchrotron high-pressure powder X-ray diffraction Rietveld refinement, which was carried out on NFS-H powder (inset: SEM image. The color of light blue was obtained via additional manipulation in the original SEM image for better color contrast). **c, d** The Na⁺ migration channels (blue-green isosurface) revealed by BVSE calculation for $Na_{2.26}Fe_{1.87}(SO_4)_3$ and $Na_6Fe(SO_4)_4$ crystals, respectively. **e** Na⁺ migration barriers along different directions. **f** Energy profiles of path Na2–Na2 along 1D direction.

directions in $Na_{2.26}Fe_{1.87}(SO_4)_3$ and $Na_6Fe(SO_4)_4$ crystals, respectively (Fig. 1e). Notably, except for the richer 3D Na⁺ diffusion channels inside the crystals, $Na_6Fe(SO_4)_4$ featured a lower barrier for Na⁺ migration within these pathways in different directions. With the 1D migration pathway as an example, the Na2–Na2 transfer displayed a lower energy barrier of 1.29 eV in the $Na_6Fe(SO_4)_4$ phase, compared with that in the $Na_{2.26}Fe_{1.87}(SO_4)_3$ phase (3.52 eV, Fig. 1f). The calculation result from another calculation software (softBV) displayed similar trend but lower barrier values (Supplementary Fig. 8 and Note 1). Furthermore, the determined diffusion coefficients of the as-prepared $Na_6Fe(SO_4)_4$ phase ranged from $10^{-10.54}$ to $10^{-7.21}$ cm² s⁻¹ at 25 °C (Supplementary Fig. 9 and Note 2), which were higher than those of pure alluaudite-type $Na_{2+2x}Fe_{2-x}(SO_4)_3$[26]. Therefore, the $Na_6Fe(SO_4)_4$ phase, which has been widely regarded as an impurity during the synthesis of $Na_{2+2x}Fe_{2-x}(SO_4)_3$ materials[29], may act as a superionic conductor in bulk materials, which enhances the ionic conductivity and reaction kinetics of the $Na_{2.26}Fe_{1.87}(SO_4)_3$ phase.

The morphology and heterostructure of NFS-H were revealed by high-angle annular dark field scanning transmission electron microscope (HAADF-STEM) images. By intense ultrasonic treatment on NFS-H, the separated nanoparticles with a trapezoidal-like morphology were obtained (Supplementary Fig. 10), in which a lattice fringe spacing of 0.361 nm pointed to the (13-1) plane of $Na_{2.26}Fe_{1.87}(SO_4)_3$. Meanwhile, a well-designed carbon network was observed around the nanoparticles. Based on thermal gravimetric analysis, the carbon content in NFS-H reached 12.64 wt%, which indicated a high-degree graphitization that was beneficial for electronic conductivity[32]. The HAADF-STEM images further identified the fine crystal fringes spread inside the NFS-H particle, following the highly crystalline feature

revealed by the XRD pattern (Fig. 2a). The lattice fringes of 0.277 and 0.308 nm matched well the (240) and (11-2) planes of $Na_{2.26}Fe_{1.87}(SO_4)_3$, respectively, and abundant (11-2) planes were exposed at the periphery of particles. However, other crystal planes such as the (240), (13-1), and (200), were also observed (Supplementary Fig. 11 and Note 3). Furthermore, the heterostructure between $Na_{2.26}Fe_{1.87}(SO_4)_3$ and $Na_6Fe(SO_4)_4$ with close contact was visible and with evident grain boundaries, in which the lattice fringes of 0.341 and 0.358 nm matched well the (21-2) plane of $Na_6Fe(SO_4)_4$ and the (13-1) plane of $Na_{2.26}Fe_{1.87}(SO_4)_3$, respectively (Fig. 2b). The corresponding "bright-bright-dark" arrangements resolved the two types of bright spot arrangement (Fig. 2c), which was also characterized by scan lines and fast Fourier-transform (FFT) patterns (Supplementary Fig. 12). Furthermore, the element mappings indicated the coexistence and uniform distribution of Na, Fe, S, and O in NFS-H particles, which is in good agreement with their tight combination. (Supplementary Fig. 3). The atomic ratio of Na:Fe (1.42:1), which was determined by energy-dispersive X-ray spectroscopy (EDS) measurements (Supplementary Fig. 13), was close to the value (1.47:1) calculated from the refined XRD result, which supports the analysis of composition contents. Nevertheless, the difference between the $Na_{2.26}Fe_{1.87}(SO_4)_3$ and $Na_6Fe(SO_4)_4$ phases cannot be determined by Fourier-transform infrared spectroscopy (FTIR) and X-ray photoelectron spectroscopy (XPS, Supplementary Fig. 14) due to the similar physicochemical properties of their elements and groups.

To reveal the structural evolution of heterostructured phases inside the cathode bulk, we studied the ex situ HAADF-STEM images at different electrochemical states (Fig. 2d, e). At a charge of 4.5 V (SOC, 100%), the lattice spacing of (21-2) plane in $Na_6Fe(SO_4)_4$ decreased and

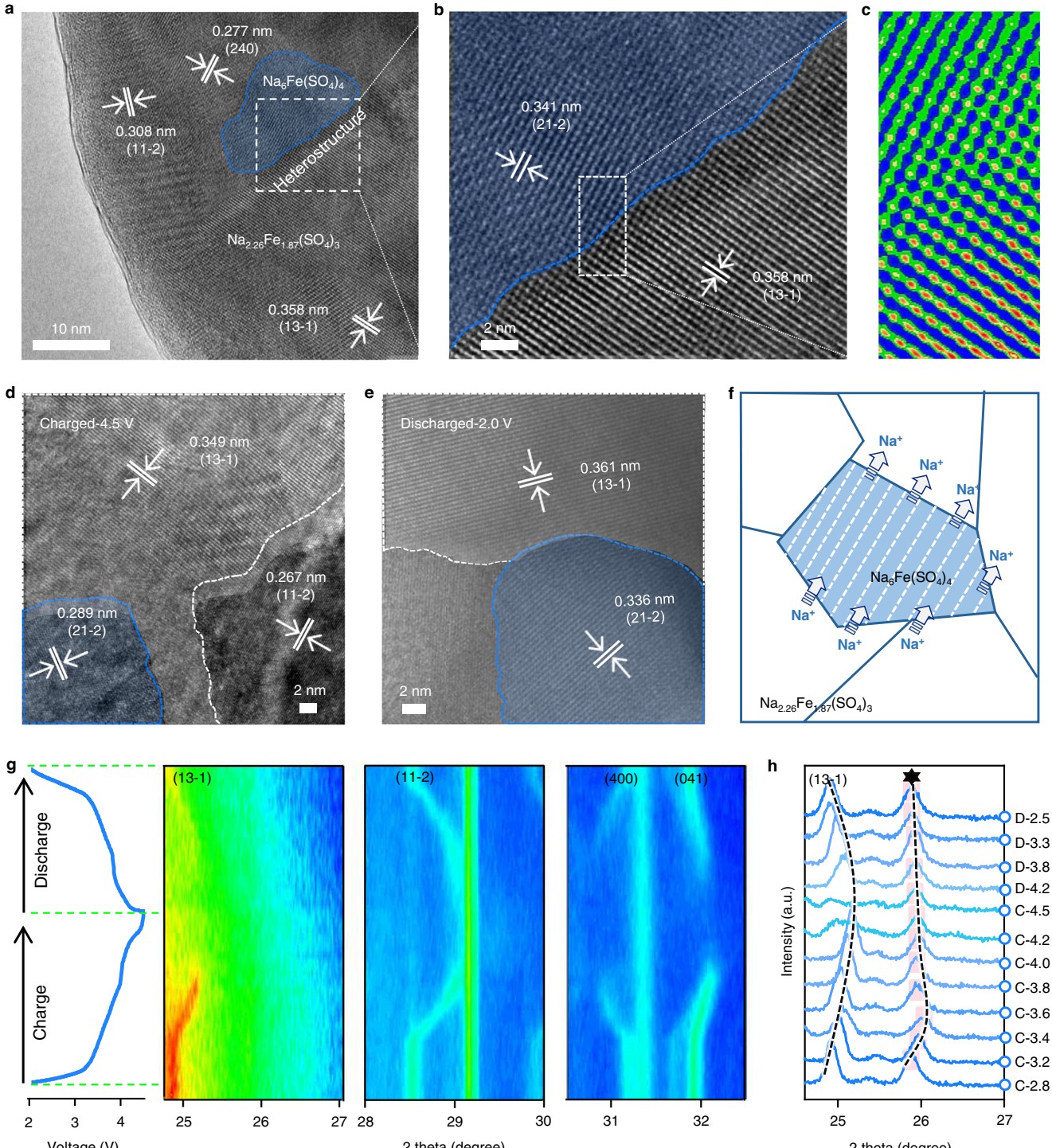

**Fig. 2 | Structure evolution of NFS-H during the electrochemical process.** HAADF-STEM images at **a**, **b** pristine state, **c** enlarged image of **b**, **d** charged 4.5 V, and **e** discharged 2.0 V upon initial cycle. The blue shadows represent $Na_6Fe(SO_4)_4$ phase materials. Two Fe arrangements are identified, which are characterized by bright Fe atoms from well-known high atomic numbers. The top-left region with a sparse Fe arrangement corresponds to Fe-less $Na_6Fe(SO_4)_4$, and the bottom-right region with a dense Fe arrangement corresponds to Fe-rich $Na_{2.26}Fe_{1.87}(SO_4)_3$. **f** Scheme of $Na^+$ transfer process. **g** In situ XRD patterns. **h** Ex situ XRD patterns at various voltages. The terms of c and d represent the charging and discharging process, respectively, and the values is the voltage at which the Na||NFS-H coin cells were stopped, disconnected, and disassembled for sample collection. The six-point star symbol stands for the (21-2) plane of $Na_6Fe(SO_4)_4$.

then returned to nearly the original value at a discharge of 2.0 V (SOC, 0%). This variation indicates that the $Na_6Fe(SO_4)_4$ phase involved electrochemical reactions during the sodiation process (Fig. 2f). The ex situ and in situ XRD patterns further demonstrated their structural evolution (Fig. 2g, h). During the charging process of $Na_6Fe(SO_4)_4$, a minor shift to a higher degree marked the emergence of the desodiated

$Na_{6-x}Fe(SO_4)_4$ phase, which continued the shift trend until 3.5 V. Afterward, characteristic reflections of desodiated $Na_{2.26-x}Fe_{1.87}(SO_4)_3$ appeared with the canting of lattice planes, including the (13-1), (11-2), (400) and (041) planes. Notable, at the beginning of $Na_{2.26}Fe_{1.87}(SO_4)_3$ desodiation, the diffraction peak of $Na_{6-x}Fe(SO_4)_4$ shifted toward a lower degree during the charging process, which indicated the partial

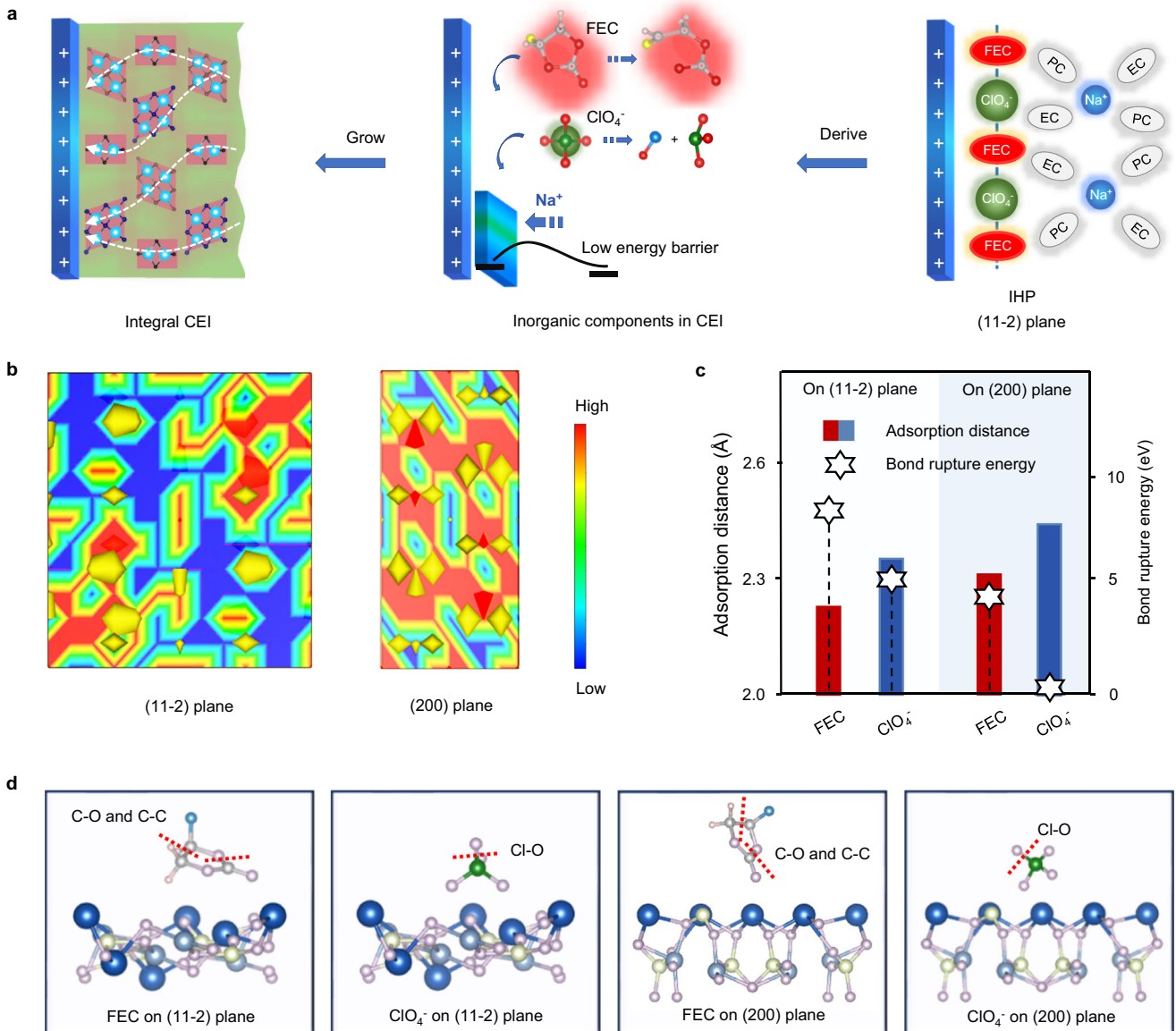

**Fig. 3 | Interface chemistry on NFS-H-based positive electrode. a** Scheme of interfacial adsorption and CEI formation. The blue flat column represents the positive active material, and the ball on the right represents the ions and solvent molecules in the electrolyte. The small flat panel in the middle represents the CEI produced by the decomposition of the electrolyte. The green and red areas on the left represent organic and inorganic species within the CEI, respectively, and the dashed lines in white represent the transport paths of sodium ions within the CEI. **b** Electron density distributions of (11-2) and (200) planes of $Na_{2.26}Fe_{1.87}(SO_4)_3$, respectively. **c** Interfacial adsorption and decomposed behaviors of electrolyte components (FEC and $ClO_4^-$) on different cathode crystal planes with corresponding adsorption distances and bond rupture energies. **d** Calculated models. Blue, dark gray, light yellow, lilac, gray, light pink, green, and water balls represent Na, Fe, S, O, C, H, Cl, and F atoms, respectively. The sticks represent the chemical bonds between atoms, and the red dotted line represents the positions where the bond breaks.

Na$^+$ sodiation from $Na_{2.26}Fe_{1.87}(SO_4)_3$ to $Na_{6-x}Fe(SO_4)_4$. During the subsequent electrochemical process, the peaks retained a gentle evolution trend and finally returned to the initial peak position upon the complete sodiation of $Na_{2.26-x}Fe_{1.87}(SO_4)_3$. The ex situ XPS spectra also suggested a similar change in the chemistry and reaction reversibility, which confirmed the solid-solution transition mechanism (Supplementary Fig. 15)[33]. Rich Na$^+$ transport channels were maintained inside the partially desodiated $Na_{6-x}Fe(SO_4)_4$, as demonstrated by BVSE calculation (Supplementary Fig. 16). The results showed that, the heterostructured $Na_6Fe(SO_4)_4$ phase and its partially desodiated phase played a crucial role in Na$^+$ conduction during the (de)sodiation process of $Na_{2.26}Fe_{1.87}(SO_4)_3$ and finally contributed to the ionic transfer and reaction reversibility of NHS-H cathode by utilizing high-throughput ion channels and low-energy barriers.

Ionic-transfer kinetics was strongly associated with the interphase structure and components in the bulk structures, and highly active cathode surfaces produce poorly ionically conductive and thick CEI, which can hamper the fast and reversible Na$^+$ transfer across the interface (Fig. 3a)[34]. To reveal the effect of the cathode surface on the formation of optimized CEI, we conducted density functional theory (DFT) calculation on the (11-2) and (200) planes of $Na_{2.26}Fe_{1.87}(SO_4)_3$ due to their peak variation during the intermediate-involved synthesis. Meanwhile, considering that the inorganic species within CEI, which was mainly derived from the fluoroethylene carbonate (FEC) electrolyte additive and ($ClO_4^-$) salt-derived anions, largely determine the uniformity, robustness, and ionic conductivity, their adsorption and decomposition behaviors in the different crystal planes were studied. Compared with the (200) plane, the (11-2) plane featured less electron

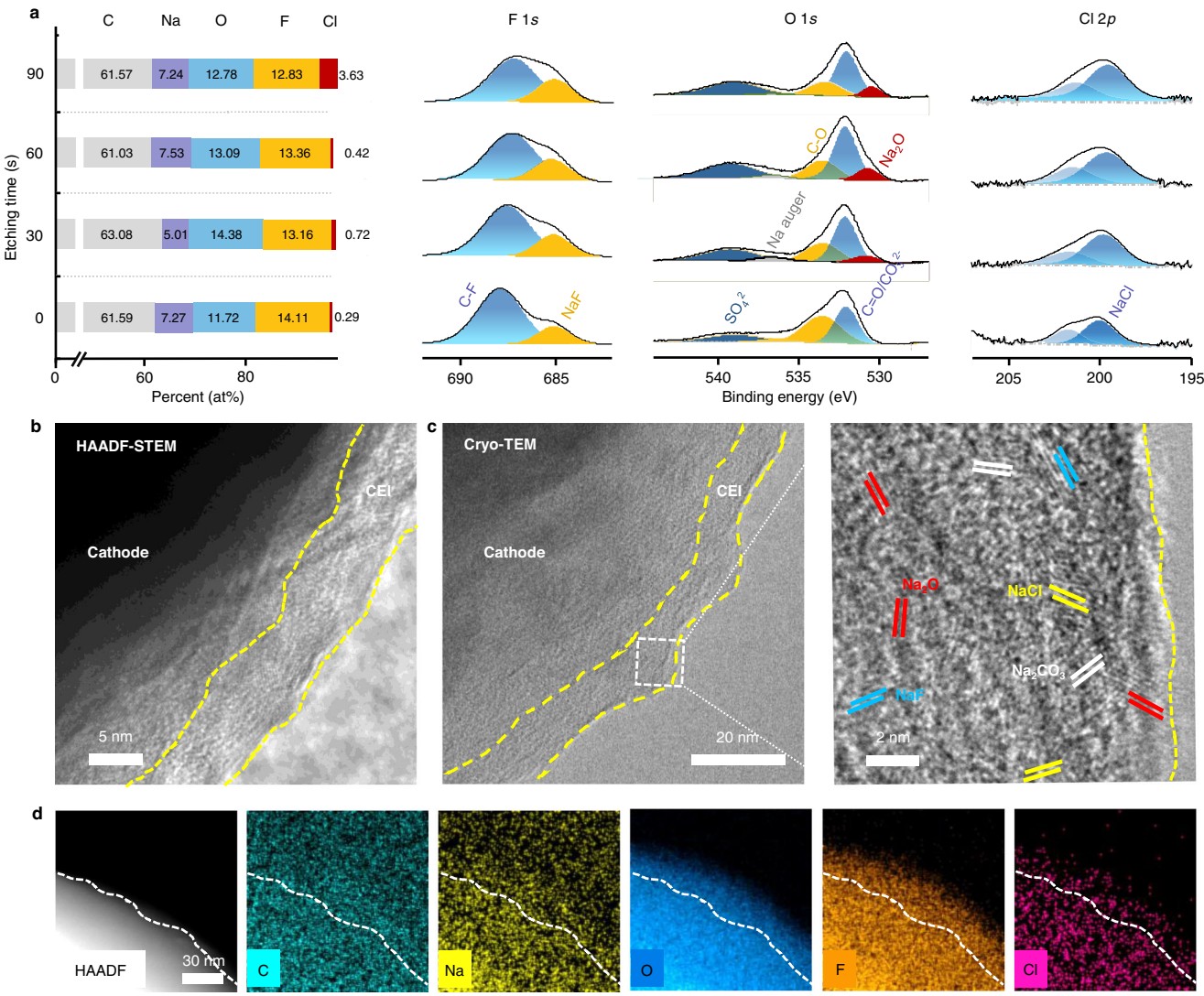

**Fig. 4 | Ex situ physicochemical characterizations of the NFS-H-based positive electrodes.** The electrodes were tested in Na||NFS-H coin cells at 12 mA g⁻¹, and the cell was stopped at 4.5 V during the first charge cycle before disassembly. **a** In-depth element distributions and high-resolution XPS spectra of O 1s, F 1s, Cl 2p. **b** HAADF-STEM image. **c** Cryo-TEM images. **d** STEM-EDS mappings.

density and stronger nucleophilicity due to the loose atom arrangement (Fig. 3b). In terms of electrolyte nucleophilic reaction on the electrode surface, the high electron density of crystal planes resulted in a stronger electron-donating capability, which decreased the barrier in the decomposition of the non-aqueous electrolyte solution[19]. Thus, the FEC and ClO₄⁻ on the (11-2) plane displayed higher bond rupture energies (C−O and C−C bonds; Cl−O bond, respectively) compared with those on the (200) plane, which implied a stronger oxidation resistance (Fig. 3c, d). Furthermore, the FEC and ClO₄⁻ with strong polar groups (F and O) featured stronger adsorption behaviors on less-electron-dense (11-2) plane with smaller adsorption distances, which indicated the improved access to the cathode surface and contributed to the formation of FEC and ClO₄⁻-rich electric double layer (EDL)[35,36]. Once the cathode reached an appropriate positive potential, the nearest FEC and ClO₄⁻ preferentially transferred electrons to the cathode and underwent oxidative decomposition, regardless of the high oxidation resistance. Accordingly, the CEI formed on the cathode with the more exposed (11-2) plane was expected to feature-rich inorganic species and ensure sufficient compactness and ionic conductivity[37]. Meanwhile, the enhanced oxidation resistance of the electrolyte can strengthen further the stability of the cathode/electrolyte interface[38]. These findings firmly indicate the capability of the

cathode's surface properties to regulate the interfacial behavior of electrolytes. The specific exposed crystal plane of the cathode material (e.g., (11-2) plane of Na₂.₂₆Fe₁.₈₇(SO₄)₃) favored the strong adsorption of several electrolyte components (e.g., FEC and ClO₄⁻) and effective construction of the target EDL, which ensured a distinctive inorganic-rich CEI to stabilize the cathode/electrolyte interface at high voltages.

CEI, as the passivation layer on the cathode surface, determines the ionic transfer and compatibility of the cathode/electrolyte interface, especially at high voltages. From the ex situ XPS full-spectrum measurements of the positive electrode (Fig. 4a), the element contents (Na, O, C, F, and Cl) changed in the etching process, with the F and Cl preserved after 90 s etching, which indicates the presence of residual CEI on the cathode surface. In addition, the highs contents of F (14.11 at.%) and Cl (3.63 at.%) implied the preferred decomposition of FEC and ClO₄⁻ anions. The C 1s spectrum showed the C-C/C-H, C-O, and -OCO₂- peaks, which indicated standard CEI components, such as sodium alkyl carbonate, sodium carbonate (Na₂CO₃), and polyester (Supplementary Fig. 17)[39,40]. Similarly, the O 1s spectrum exhibited strong peaks of C=O, Na auger, C-O/O-H, and Na₂O. Strong NaF and C-F peaks in the F 1s spectrum were derived from the oxidation of FEC, and NaCl was the characteristic decomposition product of ClO₄⁻. Through Ar⁺ etching treatment, the peak intensities, including C-C/C-H, C-F, and

-OCO$_2$, gradually decreased, which illustrated the principal distributions of organic species (C-C/C=C, C-F species, RCOO$_2$, and polyesters) in the outer CEI. Inorganic Na$_2$O appeared after 30 s etching, whereas NaF and NaCl were distributed throughout the whole depth, which implied the rich inorganic components in CEI[41].

The ex situ HAADF-STEM images of the positive electrodes demonstrated a uniform and integral CEI, with a thickness of 5−7 nm, that fully covered the cathode surface after the initial cycle (Fig. 4b). However, the CEI is susceptible to damage by beam[42]. Cryo-transmission electron microscopy (cryo-TEM) images further determined that the thickness of CEI was 10−14 nm (Fig. 4c). The organic species corresponding to amorphous regions ensured a fully passivated interface, and the high-content inorganic components (NaF, Na$_2$O, and NaCl) corresponding to areas filled with various lattice fringes provided rapid Na$^+$ transfer pathways across the interface and remarkable interface robustness. STEM-EDS images further indicated the intensive distribution of F, Cl, Na, and O on the cathode surface (Fig. 4d). It effectively facilitated the Na$^+$ transfer across the interface and interfacial robustness to maintain its integrity during the long-term cycling (Supplementary Fig. 18), which was also demonstrated by the low activation energy (48.6 kJ mol$^{-1}$) calculated from the analysis of electrochemical impedance spectroscopy (EIS) measurements at various temperatures (Supplementary Fig. 19 and Supplementary Table 5). Considering the contribution of inorganic species to the ionic conductivity and robustness of CEI[43], an inorganic-rich CEI well protected the cathode and minimized adverse interfacial reactions, which resulted in a stable interface and rapid Na$^+$ transfer during the long-term cycling of the cell.

Benefiting from the design of continuous Na$^+$ channels (Supplementary Fig. 20), the NFS-H cathode displayed enhanced ion transfer kinetics and reaction reversibility. The voltage profile of the Na||NFS-H coin cell showed an initial discharge capacity of 101.3 mAh g$^{-1}$ and a discharge cell voltage of 3.75 V at a specific current of 6 mA g$^{-1}$ (Supplementary Fig. 21 and Note 4). By contrast, the NFS-L with a less ionic-conducting phase featured a larger hysteresis and a lower initial discharge capacity (89.7 mAh g$^{-1}$) at the same specific current. This finding indicated the improved reaction kinetics of the ionic-conductive Na$_6$Fe(SO$_4$)$_4$ phase, which facilitated the decreased electrochemical polarization. Moreover, the low sodium-ion storage capacity of Na$_6$Fe(SO$_4$)$_4$ cathode might imply poor Fe$^{2+}$/Fe$^{3+}$ redox reversibility (Supplementary Fig. 9)[29]. Thus, the sodium-ion storage capacity of NFS-H was not entirely attributed to the de/intercalation accompanied by Fe$^{2+}$/Fe$^{3+}$ redox reaction in Na$_{2.26}$Fe$_{1.87}$(SO$_4$)$_3$, but also to minor contributions of the Na$_6$Fe(SO$_4$)$_4$ phase. At higher currents (600 mA g$^{-1}$), the Na metal cell with the NFS-H-based positive electrode showed a discernable voltage inflexion around 3.0 V in the discharge curves (Supplementary Fig. 21), which may be associated with the Fe$^{2+}$/Fe$^{3+}$ redox couple of the Na$_6$Fe(SO$_4$)$_4$ phase in NFS-H[29]. The pairs of weak peaks at 2.89/3.05 and 2.62/2.83 V in the CV curves of Na||NFS-H and Na||NFS-L cells were possibly caused by trace Na$_6$Fe(SO$_4$)$_4$ phase (Supplementary Fig. 21)[39]. The sloping voltage curve over the entire range of Na composition in Na||NFS-H and Na||NFS-L cells suggests the (de)sodiation of Na$^+$ and a single-phase homogeneous reaction mechanism involving minimal volume change (Supplementary Fig. 21)[26,44].

Na metal cells with NFS-H-based positive electrodes displayed excellent rate performance, with discharged capacities of 101.3, 99.7, 98.0, 95.0, 92.1, 88.6, 82.1, and 73.5 mAh g$^{-1}$ at 6, 12, 24, 60, 120, 240, 600, and 1200 mA g$^{-1}$, respectively (Fig. 5a). When the specific current returned to 12 mA g$^{-1}$, it retained a high discharged capacity of 97.4 mAh g$^{-1}$, which was higher than those obtained for the Na||NFS-L cells. The charge−discharge curves of Na||NFS-H cells at different specific currents exhibited similar voltage plateau and good stability, which indicated the relatively low polarization at high specific currents (Supplementary Fig. 21). Compared with the reported Fe-based sulfate

cathodes, the as-prepared NFS-H exhibited excellent Na$^+$ storage capacity at various rates in laboratory-scale Na metal cell configuration (Supplementary Fig. 22). The CV curves at different scan rates indicated a capacitive-dominated kinetic process for Na metal cells with the NFS-H cathode (Supplementary Fig. 22 and Note 5)[45]. In addition, the galvanostatic intermittent titration technique (GITT) test further determined the Na$^+$ diffusion coefficients (D$_{Na+}$) of 10$^{-12.44}$ to 10$^{-10.52}$ cm$^2$ s$^{-1}$ (Fig. 5b, c), which are comparable to those of its homologs and NASICON-type materials (Supplementary Table 6); these findings highlight the effective improvement of heterostructural Na$_6$Fe(SO$_4$)$_4$ on the ionic transfer of Na$_{2.26}$Fe$_{1.87}$(SO$_4$)$_3$.

The Na||NFS-H coin cell demonstrated a capacity retention of 80.69% at 60 mA g$^{-1}$ after 1300 cycles (127 days, Fig. 5d). Furthermore, the capacity fading mechanisms were discussed based on coulombic efficiency (CE), EIS measurements, and direct-current (DC) resistances. First, the Na metal coin cells with NFS-L displayed a lower initial CE (72.2%) and average CE (98.66%) than Na||NFS-H coin cells (82.3 and 99.59%, respectively). Meanwhile, the interfacial resistance increased by 31.3 and 55.1% during the initial 50 cycles in Na metal coin cells using NFS-H and NFS-L, respectively (Supplementary Fig. 23). In addition, at the 1200th cycle, the discharged DC resistances of Na metal coin cells using NFS-H and NFS-L showed evident increases of 154.6 and 552.8%, respectively. Therefore, capacity fading can be ascribed to continuous interfacial side reactions and the thickening of CEI (Supplementary Fig. 18 and Note 6). On the other hand, the insertion/extraction of Na$^+$ leads to changes in the molar volume of Na$_{2.26}$Fe$_{1.87}$(SO$_4$)$_3$ and Na$_6$Fe(SO$_4$)$_4$ phases, which may induce mechanical stress and strain to their crystals. Thus, the detachment between the two phases was possible in local regions, which may damage the ion-conduction networks inside the electrodes and promote diffusion impedance. The introduced ionic-conducting Na$_6$Fe(SO$_4$)$_4$ phase, which built sufficient ion diffusion channels in NFS-H, can maintain the reversible (de) insertion reaction and capacity well. As a result, at 600 mA g$^{-1}$, the Na|| NFS-H coin cell maintained 65.32% of the initial capacity after 5700 cycles without sudden capacity loss or large fluctuation in CEs (Fig. 5e)[46]. High average CEs (99.79% within 5700 cycles) highlight the reversibility of the solid-phase reaction of heterostructured phases and stable cathode/electrolyte interface for rapid and stable Na$^+$ transfer, which enabled the Na||NFS-H coin cells with excellent cycling-stability compared with the state-of-the-art Na-based laboratory-scale cells using Fe-based positive electrode active materials (Supplementary Table 7).

To verify the feasibility of an NFS-H-based positive electrode for practical applications, we assembled an all-iron-sulfur sodium-ion cell with presodiated FeS-based negative electrode (Fig. 6a, b). The Na-ion laboratory-scale pouch cell showed a charge−discharge capacity of 83.9/81.1 mAh g$^{-1}$ based on the mass of the positive electrode active material (Fig. 6c). After 40 cycles at 24 mA g$^{-1}$, the Na-ion cell retained 96.69% of its initial capacity with a stable average discharge voltage of 2.35 V (Fig. 6d). Meanwhile, the Na-ion cell exhibited good rate performance with a specific power of 486.5 W kg$^{-1}$ at 240 mA g$^{-1}$ (Supplementary Fig. 24). Compared with other reported laboratory-scale Na-ion cells, the all-iron-sulfur cell delivered a competitive specific energy of 168.2 Wh kg$^{-1}$ (total mass of the positive and negative electrodes)[13,47–49]. The cell system mainly consisted of resource-abundant and nontoxic elements (Fe, S, and O) to avoid the dilemma of the utilization of raw materials and cost surge during potential large-scale applications (Fig. 6e). Furthermore, the system contained mutual and element compositions in positive and negative electrodes, which suggests the possibility of effective electrode recycling (Fig. 6f and Supplementary Tables 8, 9)[50].

In summary, an ionic-transfer-enhanced iron-based sulfate with an efficient carbon network for electron conductivity was fabricated via co-precipitation and subsequent calcination process as high-voltage positive electrode active materials for non-aqueous

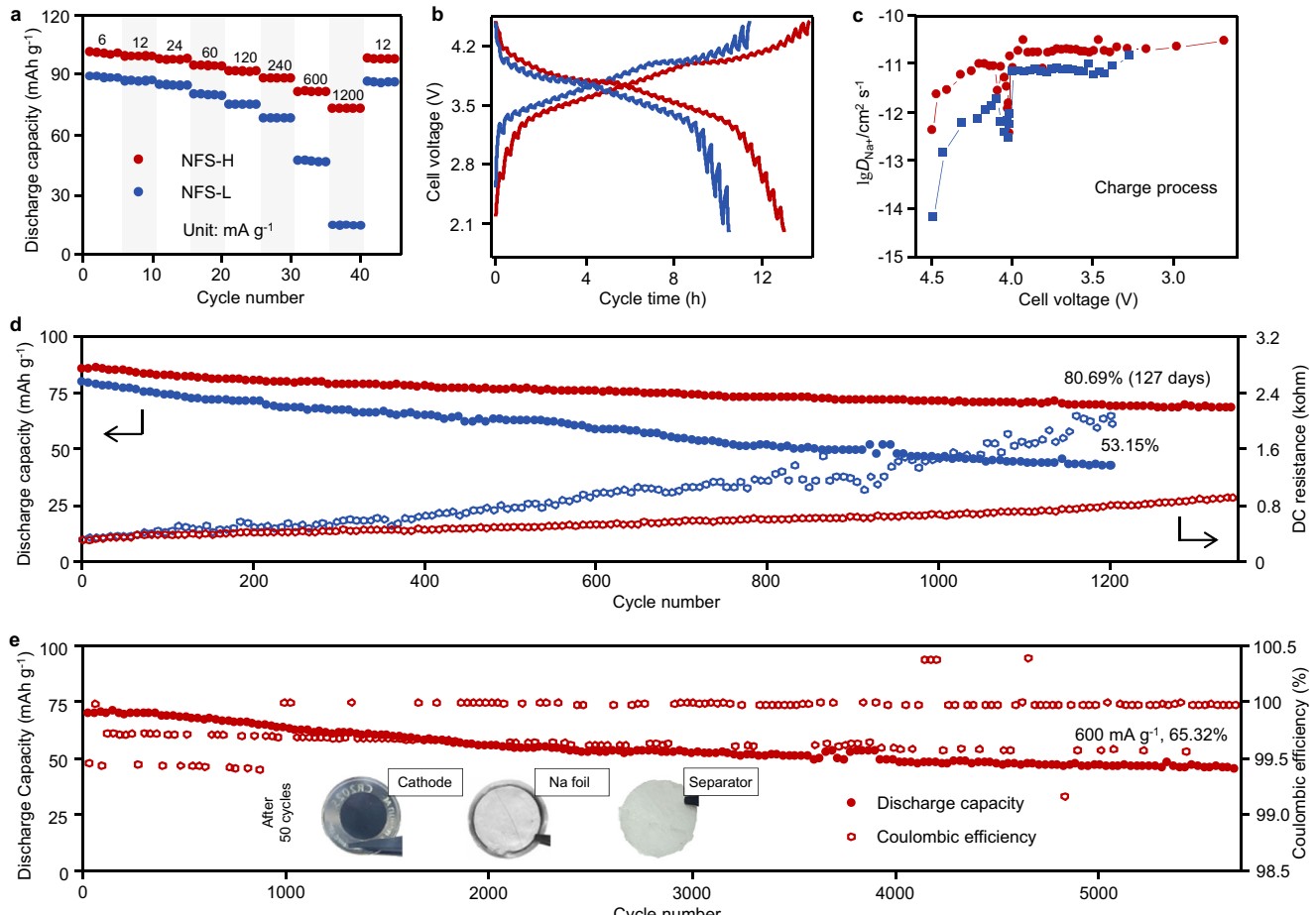

**Fig. 5 | Electrochemical performance of NFS-H- and NFS-L- based electrodes in non-aqueous Na metal coin cell configuration. a** Rate capability. **b** GITT curve. **c** The calculated Na$^+$ diffusion coefficients. **d** Cycling performance and DC internal resistance at 60 mA g$^{-1}$. **e** Long cycling performance and Coulombic efficiency at 600 mA g$^{-1}$, inset: photos of disassembled battery. The Na||NFS-H coin cells were stopped and disassembled after 50 cycles at discharge state (2.0 V) to collect the photographic pictures disclosed in the insets.

Na-based batteries. The presence of heterostructure of Na$_6$Fe(SO$_4$)$_4$ phase (10.5 wt%) inside the bulk of particles was confirmed by HAADF images. BVSE calculations demonstrated abundant 3D Na$^+$-diffusion channels and low barriers of Na$_6$Fe(SO$_4$)$_4$, which enhanced the ionic conductivity of the cathode. The low-electron-density (11-2) crystal plane of the main phase (Na$_{2.26}$Fe$_{1.87}$(SO$_4$)$_3$) induced the strong adsorption of FEC and ClO$_4^-$ for the construction of a passivating EDL, which derived an integrated and inorganic-rich CEI. These speculations were supported by ex situ cryo-TEM and XPS measurements and analyses and DFT calculations. The inorganic components of the CEI such as NaF, Na$_2$O, and NaCl, provided efficient Na$^+$-diffusion channels and contributed to high Na$^+$ diffusion coefficients (D$_{Na^+}$, 10$^{-12.44}$–10$^{-10.52}$ cm$^2$ s$^{-1}$). Meanwhile, the uniform CEI effectively prevented the decomposition of electrolytes at high voltages (4.5 V). As a result, the Na metal cell employing the composite iron sulfate-based positive electrode delivered an initial discharge capacity of 101.3 mAh g$^{-1}$ at 6 mA g$^{-1}$ (with an average discharge cell voltage of about 3.75 V), good rate performance (73.5 mAh g$^{-1}$, 1200 mA g$^{-1}$) and cycling-stability (80.69 and 65.32% capacity retention after 1300 cycles at 60 mA g$^{-1}$ and 5700 cycles at 600 mA g$^{-1}$, respectively). Moreover, an all-iron-sulfide sodium-ion laboratory-scale pouch cell assembled with presodiated FeS-based negative electrode exhibited earth-abundant element compositions and calculated initial specific energy density of 168.2 Wh kg$^{-1}$ (based on the mass of the positive and negative electrodes) at 24 mA g$^{-1}$ and possible electrode recycling.

## Methods

### Materials
Iron sulfate heptahydrate (FeSO$_4$·7H$_2$O, ≥99.0%), anhydrous sodium sulfate (Na$_2$SO$_4$, ≥99.0%), ascorbic acid (C$_6$H$_6$O$_6$, ≥99.7%), and citric acid monohydrate (C$_6$H$_{10}$O$_8$, ≥99.5%) were bought from Sinopharm Chemical Reagent Co., Ltd. without further purification. Graphene oxide (GO) sheet powder was bought from Shenghui's New carbon material store on the Taobao website. Ethylene carbonate (EC, 99.95%), propylene carbonate (PC, 99.99%), FEC (99.9%), and NaClO$_4$ salt (99.99%) was bought from the Aladdin website. Conductive carbon (super P, TIMICAL, >98.5% purity, D$_{50}$ = 40 nm), polyvinylidene fluoride (PVDF, Solvay5130), Whatman GF/D microfiber filter paper (~200-μm-thick, average pores size of 2.7 μm, Cytiva), Al foil (Al >99.999%, 20 μm), and copper foil (9 μm) were bought from Canrd New Energy Technology Co., Ltd. Na metal (99.7%) was bought from Beijing Yinokai Technology Co., Ltd.

### Material preparation
In a typical procedure, a solution was obtained by dispersing 20 mg GO and 10 mL ethylene glycol (EG) in 20 mL deionized water followed by 30 min of ultrasonic dispersion (100 W, 25 °C, KQ-250DS, Kunshan Ultrasonic Instruments Co. Ltd). FeSO$_4$·7H$_2$O (4 mmol), Na$_2$SO$_4$ (4 mmol), ascorbic acid (0.02 g), and citric acid monohydrate (0.2 g) were dissolved in the above solution and stirred for 1 h at 25 °C. Subsequently, 40 mL isopropanol was added dropwise to the above solution and stirred for 1 h at a low temperature (<5 °C), followed by

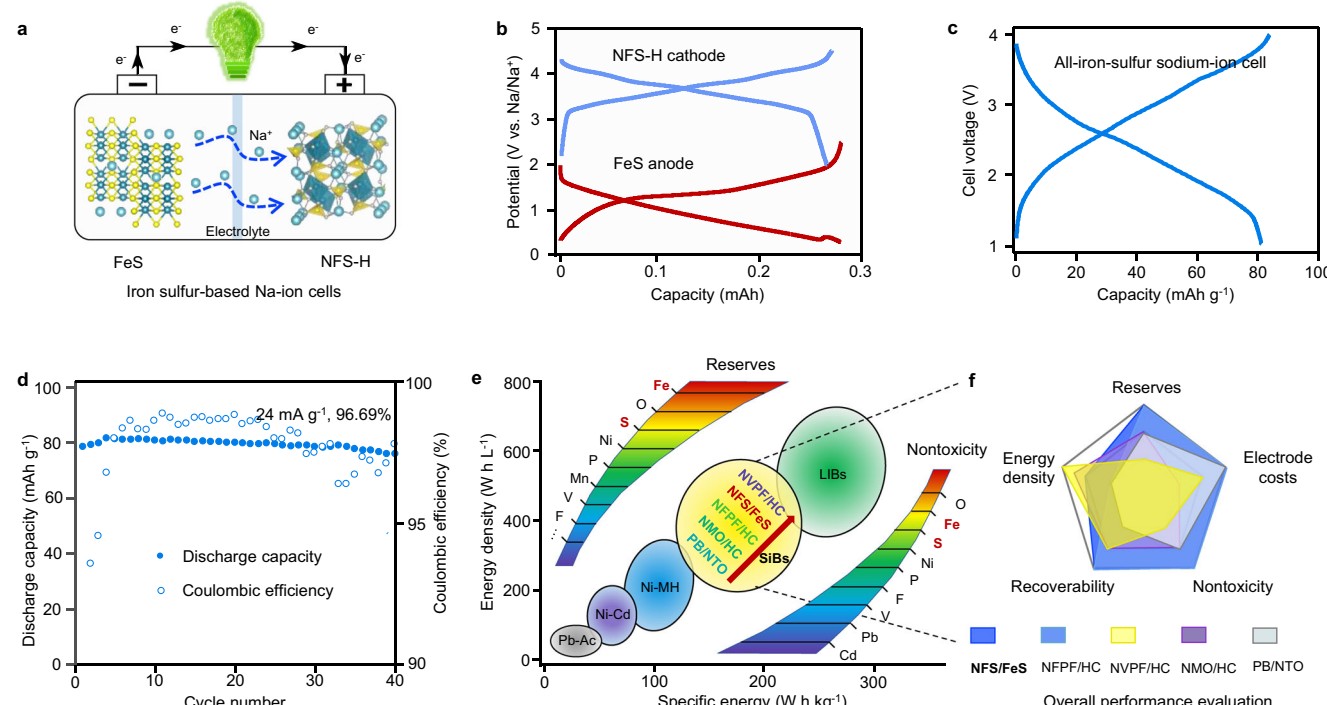

**Fig. 6 | Electrochemical performance of assembled laboratory-scale iron sulfur-based Na-ion cells. a** Schematic representation of the FeS||NFS-H cell. **b** Charge–discharge curves of Na||NFS-H and Na||FeS cells at fourth cycle, respectively. **c** Fourth charge–discharge voltage profiles of the FeS||NFS-H cell cycled at 24 mA g$^{-1}$. **d** FeS||NFS-H cell cycling performance. **e**, **f** The overall performance evaluation with typical storage energy technologies and battery systems in SiBs. The color gradient from blue to red represents improved elemental abundances on earth and the nontoxic property of elements. The comprehensive evaluation of various sodium-ion battery systems and calculation parameters of electrode costs are presented in Supplementary Tables 8 and 9, respectively. The terms NFPF, HC, NVPF, NMO, PB, and NTO represent the electrode materials of Na$_2$FePO$_4$F, hard carbon, Na$_3$(VOPO$_4$)$_2$F, Na$_{0.76}$Cu$_{0.22}$Fe$_{0.30}$Mn$_{0.48}$O$_2$, Na$_{1.64}$Ni[Fe(CN)$_6$]$_{0.92}$·1.83H$_2$O, and NaTi$_2$(PO$_4$)$_3$, respectively.

centrifugation (8000 rpm for 3 min, TG16, Shanghai Lu Xiangyi Centrifuge Instrument Co., Ltd) and freeze-drying (LGJ-10N/C, Beijing Yaxing Yike Technology Development Co. Ltd) overnight to obtain the precursor. After full grinding in the air by a human operator using a mortar and pestle (agate material), the precursor was sintered at 350 °C for 12 h in argon at a heating rate of 2 °C/min. The final product was ground (in the air by a human operator using a mortar and pestle) again and denoted as NFS-H. By comparison, a heterostructure-less material was prepared without the addition of EG by promoting single-phase precipitated FeSO$_4$·4H$_2$O, which was denoted as NFS-L.

For the preparation of Na$_6$Fe(SO$_4$)$_4$, the mixture of anhydrous NaSO$_4$, FeSO$_4$, and 10 wt% carbon black materials (super P) with a molar ratio of 3:1 was placed in a tank (corundum tank, 250 mL, 95% ZrO$_2$ ball) in an environment filled with Ar gas at 25 °C, and ball milling was carried out by a high-speed planetary ball mill (ZW-Q250, Kunshan Zhuowei Machinery Technology Co. Ltd). This process lasted for 6 h at the ball-mill speed of 500 rpm/min. The obtained material was then calcined in a tubular furnace at 350 °C for 12 h at a heating rate of 1 °C/min in Ar gas. Finally, Na$_6$Fe(SO$_4$)$_4$ was obtained.

For the preparation of FeS[51], 10.92 g C$_6$H$_8$O$_7$·H$_2$O and 8.82 g FeSO$_4$·7H$_2$O were dissolved in 50 mL deionized water. After sufficient stirring, the solution was frozen using liquid nitrogen, and then freeze-dried (LGJ-10N/C, Beijing Yaxing Yike Technology Development Co. Ltd) under vacuum for 36 h. The obtained precursor was further heated at 800 °C for 6 h in Ar at the rate of 5 °C min$^{-1}$, and the FeS material was obtained.

## Materials characterization

Synchrotron high-pressure powder XRD was carried out at the BL14B1 beamline of Shanghai Synchrotron Radiation Facility using X-rays with a wavelength of 0.6887 Å. The morphologies were systematically characterized by SEM (ZEISS Merlin Compact) coupled with energy-dispersive X-ray (EDX) analysis. TEM tests were carried out using an FEI Titan G2 60-300 microscope at 300 kV, and HAADF-STEM images were measured by JEOL JEM-ARM300F. Cyro-TEM was performed using an FEI Glacios (USA) operated at 200 kV, and equipped with an automatic injection system of frozen samples. All samples were prepared under Ar atmosphere, and TEM grids were removed from the centrifuge tube and transferred to the TEM column in the liquid N$_2$. Thermogravimetry (TG) was performed on (differential scanning calorimetry/differential thermal analysis-TG) STA 449 F3 Jupiter® in the air from room temperature to 800 °C with a temperature ramp of 10 °C min$^{-1}$. N$_2$ adsorption/desorption measurement was conducted by Brunauer–Emmett–Teller test (Micromeritics ASAP 2420). XPS was performed using Thermo Escalab 250Xi. The depth of etching for a second was ~0.26 nm based on the measurement of the standard substance TaS$_2$ by the instrument. Raman spectra were tested with a LabRAM HR800 Evolution using a 532 nm laser. FTIR spectra were determined on an FTIR spectrometer (Nicolet iS50, Thermo Scientific). In situ XRD pattern was detected in an in situ electrochemical device (Beijing Zhongyan Huanke Technology Co., Ltd), and XRD (PANalytical Empyrean) with Cu Kα radiation (λ = 1.54056 Å) and voltage of 45 kV. The working electrode was fabricated by mixing the active material, conductive carbon (super P), and polyvinylidene difluoride (PVDF) at a weight ratio of 8:1:1 in N-methyl-2-pyrrolidone (NMP) to form a slurry. The obtained slurry was coated on aluminum foil (6 μm, Qiandingli electronic technology company) and dried overnight at 120 °C in a vacuum with an electrode mass loading of 6.0–7.0 mg cm$^{-2}$. Other assembled parameters were the same as those of coin cells described in the "Electrochemical measurements" section. The ex situ XRD pattern was determined in powder XRD (BRUKER D8 ADVANCE A25)

with Cu Kα radiation ($\lambda = 1.54056$ Å) at a scan rate of 1º/min. The electrode was prepared by the same process as described in the "Electrochemical measurements" section. Before the ex situ measurements, the electrodes were washed thrice by propylene carbonate solvent and then dried for 12 h at 25 °C in an argon-filled glovebox ($O_2$ and $H_2O$ <0.1 ppm). The electrochemical process was carried out on Na||electrode coin cells, and vacuum plastic bags were used to transport the electrode samples from the Ar-filled glovebox to the equipment.

## Electrochemical measurement

The electrochemical properties were evaluated with CR2025 coin-type cells assembled in an argon-filled glovebox ($O_2$ and $H_2O$ <0.1 ppm). The working electrode was fabricated by mixing the active material, conductive carbon (super P), and PVDF at a weight ratio of 7:2:1 in NMP to form a slurry. Mixing was carried out in the air by a human operator with a mortar and pestle (agate material). The obtained slurry was coated on aluminum foil (20 μm) and dried overnight at 120 °C in a vacuum with an electrode mass loading of 1.5–2.0 mg cm$^{-2}$. The dry electrode was then punched into disks of 13 mm. Na metal coin cells were assembled using Na metal (thickness: 260–300 μm and diameter: 16 mm) as the counter/reference electrode and Whatman GF/D paper as the separator. The electrolyte was 1 M NaClO$_4$ dissolved in a mixed solution of EC/PC (1:1 by volume) with 5 wt% FEC as an electrolyte additive. The added amount of electrolyte was 120 μL for each assembled coin cell. The initial five cycles of Na metal cells were cycled at 6 mA g$^{-1}$ within 2–4.5 V. The specific capacity of cells was calculated based on the mass of active material in the positive electrode. All cells function in a constant-temperature environment (25 ± 1 °C).

The Na-ion pouch cells were assembled with an NFS-H-based positive electrode and FeS-based negative electrode and 1 M NaClO$_4$ EC/PC with 5 wt% FEC as electrolyte. The preparation of the positive electrode was the same as that for Na metal coin cells, which had a single-side coating and a diameter of 16 mm. FeS electrode was fabricated by mixing FeS material, conductive carbon (super P), and sodium carboxymethyl cellulose in water at a mass ratio of 8:1:1. The slurry was then cast on a copper foil (-10- μm-thick) and dried at 60 °C overnight, and the dry electrode was then punched into 16 mm disks (20–40 μm-thick, mass loading of 0.5–0.8 mg cm$^{-2}$). The capacity ratio of anode/cathode was 1.30–1.40. Before assembly of the full cell, the FeS-based electrodes were presodiated in Na metal coin cell configuration using 1 M NaClO$_4$ EC/PC with 5% FEC electrolyte solution within 0.01–2.5 V for three cycles. Then, the FeS electrode (at charge 2.5 V) was disassembled from the coin cell configuration and directly used as a negative electrode in Na-ion cells without additional cleaning and drying. The assembly of Na-ion cells was performed in an Ar-filled glovebox, and Whatman GF/D paper was utilized as the separator. The initial three cycles of cells were precycled at 6 mA g$^{-1}$ in the cell voltage range of 1–4 V, and external pressure was not applied to the pouch cell during battery testing.

Galvanostatic charge/discharge tests were carried out in the voltage range of 2.0–4.5 V using a NEWARE multichannel battery test system. CV tests were performed on a CHI604E (Chenhua Instrument Co., Ltd) in an oven (DH2500B incubator, Tianjin Tongli Xinda Instrument Factory), in the potential range of 2.0–4.5 V at different scanning rates. EIS measurements at various temperatures were carried out by applying a potentiostatic signal. The amplitude of the AC voltage was set at 5 mV at 100 kHz to 0.01 Hz (CHI604E), and the overall number of data points was 62. Other EIS measurements at 25 °C were performed with the potentiostatic signal, and the amplitude of the AC voltage was set at 5 mV at 100 kHz to 0.01 Hz (Autolab RRDE/RDE-2). The overall number of data points was 71. Two cells were tested for the galvanostatic charge/discharge and GITT tests, and one cell was tested for the CV at various rates and EIS at various temperatures.

## GITT calculation

GITT was carried out in a coin cell after two cycles to reach the thermal equilibrium state. The charging time was set to 600 s ($\tau$), followed by a 600 s rest process. We used the following equation to calculate the diffusion coefficient of sodium ions ($D$):

$$D = \frac{4L^2}{\pi\tau}\left(\frac{\Delta E_s}{\Delta E_\tau}\right)^2 (\tau \ll L^2/D) \tag{1}$$

where $L$ and $\tau$ represent the thickness of the coated material on the electrode and charging time per round, respectively. $\Delta E_s$ and $\Delta E_\tau$ are the voltage change of batteries during the charging (discharging) process and from static to the equilibrium state, respectively.

## Theoretical calculation methods

The migration pathways of mobile Na$^+$ were calculated based on the BVSE method and Voronoi decomposition, which were carried out on the Computing and Date Platform for Electrochemical Energy Storage Materials[30,31]. For comparison, the migration pathways of mobile Na$^+$ were also calculated using SoftBV software with a resolution of ca. 0.1 Å[352,53]. DFT calculations were performed using the Vienna ab-initio simulation package, which were based on DFT and the plane-wave pseudopotential method[54,55]. The generalized gradient approximation and Perdew–Burke–Ernzerhof exchange function were used. The plane-wave energy cutoff was set to 400 eV. The convergence criteria of energy and force calculations were set to $10^{-4}$ eV and 0.03 eV Å$^{-1}$, respectively.

## Reporting summary

Further information on research design is available in the Nature Portfolio Reporting Summary linked to this article.

# Data availability

All data supporting this study and its findings are available within the article and Supplementary Information. Additional supporting data of this study are available from the corresponding author on reasonable request. Source data are provided with this paper.

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

## Acknowledgements

The authors greatly acknowledge the financial support from the National Natural Science Foundation of China (No. 22279121), the Joint Fund of Scientific and Technological Research and Development Program of Henan Province (222301420009), Zhengzhou University, and Longzihu New Energy Laboratory. The Modern Analysis and Gene Sequencing Center of Zhengzhou University and the Institute of Microstructure and Properties of Advanced Materials at Beijing University of Technology, Shanghai Synchrotron Radiation Facility (SSRF) were appreciated for their support in tests. The authors are grateful to the National Supercomputing Centre in Zhengzhou, Henan Province Supercomputing Centre, and Computing and Date Platform for Electrochemical Energy Storage Materials for the computational support.

## Author contributions

W.C. conceived and supervised the research. J.Z. conducted the experiments, analyzed data, and prepared the manuscript. Y.Y. analyzed data and co-wrote the manuscript. X.W. and S.W. participated in synthesizing samples and conducted electrochemical performances. Y.C. conducted the paper modification. Z.Z. and P.Y. conducted the HAADF-STEM. Z.X. conducted the Cryo-TEM. All authors contributed to the results discussion and writing. J.Z. and Y.Y. contributed equally to this work.

## Competing interests

The authors declare no competing interests.
