## [Peer Review File · Nature Communications]

REVIEWER COMMENTS

Reviewer #1 (Remarks to the Author):

The manuscript by Zhang et al. reports on the role of a $\text{Na}_6\text{Fe}(\text{SO}_4)_4$ minority phase in enhancing the electrochemical performance of $\text{Na}_{2.26}\text{Fe}_{1.87}(\text{SO}_4)_3$ cathode material for Na-ion batteries. The manuscript also discussed the impact of the exposure-preferred (11-2) plane of $\text{Na}_{2.26}\text{Fe}_{1.87}(\text{SO}_4)_3$ particles in establishing a stable and high ion conducting SEI layer, which helped them achieve a high capacity of 101.3 mAh/g with an exceptional rate performance (73.5 mAh g⁻¹ at 10 C) and cyclic stability (~ 65% capacity retention after 5700 cycles). The authors have used multiple theoretical studies as well as structural & spectroscopic techniques to substantiate their findings and ascertain the formations of highly conducting Na⁺ migrations channels within the material. In view of the focus on cheaper and earth-abundant Na-ion batteries (NIBs) cathodes, there is a need to explore a broader range of materials that promise excellent rate-performance & cyclability. This is especially important as the poor Na-ions kinetics in the cathode layer are considered one of the significant challenges in the broader adoption of NIBs at the commercial scale. The results present in this manuscript are a decent addition to the research on NIB cathodes. However, the following points must be addressed in the revised manuscript before being accepted for publication in Nature Communications.

1. The authors have used theoretical calculations to substantiate the role of $\text{Na}_6\text{Fe}(\text{SO}_4)_4$ in forming highly conducting Na-ion pathways with the material. From an experimental standpoint, it is highly recommended that the authors substantiate this result through experimental means (EIS, chronoamperometry, etc.) on the as-prepared $\text{Na}_6\text{Fe}(\text{SO}_4)_4$ material to confirm its superionic conducting nature. It would be helpful to the reader and improve the quality of this manuscript if the difference in ionic conductivities of the main $\text{Na}_{2.26}\text{Fe}_{1.87}(\text{SO}_4)_3$ phase and minor $\text{Na}_6\text{Fe}(\text{SO}_4)_4$ phase is quantified. In lines 123 – 127, the authors mention the synthesis of 4.52 wt% $\text{Na}_6\text{Fe}(\text{SO}_4)_4$ phase and the metastable nature of $\text{Na}_{2.26}\text{Fe}_{1.87}(\text{SO}_4)_3$ and $\text{Na}_6\text{Fe}(\text{SO}_4)_4$ phases. Is this valid only for the co-precipitation method or in general? Nonetheless, the electrical properties of NFS-H and NFS-L can be compared. Interestingly, DCIR for both cells is quite similar (Fig. 5d) for the first 50-60 cycles!

2. XRD Analysis

a. Fig. 1(b) & Fig. S2(c) shows the XRD profiles of NFS-H with low signal-to-noise ratios. As this data is instrumental in identifying crystallographic parameters and phase fractions, the authors should obtain better-quality XRD/neutron/Synchrotron XRD data to analyze the material.

b. Also, include the error in the estimated % phase fractions.

c. Fig. 2(f) again shows poor quality data of ex-situ XRD data which makes it difficult to judge the phase formations/structural evolutions discussed by the authors.

d. For the reader's benefit, it is recommended to include details about the experimental setup used for in-situ XRD studies (cell configuration, operando/in-situ conditions, etc.)

3. Line 158: The authors have calculated the carbon content in the as-prepared material to be about 12.64 wt% of the total sample. Why was an additional 20% super-P added while preparing the cathode slurry?

4. The total capacitive contribution of NFS-H calculated from the CV curves was > 90% at a scan rate of 0.1 mV/s. The values for peaks 2 and 3 (Figure S18a) are typical for Na-ion batteries. If most of the contribution to specific capacity was from a surface-based phenomenon, what is the relevance of $\text{Na}_6\text{Fe}(\text{SO}_4)_4$ that ostensibly establish high Na-ion conduction pathways in the bulk of the material?

5. The authors should clarify the mode of existence of the $\text{Na}_6\text{Fe}(\text{SO}_4)_4$ phase, that is, if this has intergrowth within the $\text{Na}_{2.26}\text{Fe}_{1.87}(\text{SO}_4)_3$ crystallites (Figure 1a) or if it segregates at the grain boundaries of $\text{Na}_{2.26}\text{Fe}_{1.87}(\text{SO}_4)_3$ phase (Figure 2g)!

6. The SEM image of the "particle," shown in Fig. 1b inset, seems to be in the form of a polyhedron with 6 dominating faces (rhombic hexahedron). Do all these faces belong to $\{1\ 1\ -2\}$ family of the monoclinic NFS-H? Interestingly, the particle morphology for NFS-L (Figure S6) sample is drastically different, with no apparent faceting, despite both samples having the same phases.

7. Line 187, Supplementary Information: "In Figure S17a... ". S17a should be changed to S17b. The redox behavior of Fe^{2+} in the minor $\text{Na}_6\text{Fe}(\text{SO}_4)_4$ needs more discussion. The entire capacity is attributed to the de/intercalation accompanied by $\text{Fe}^{2+}/\text{Fe}^{3+}$ transformation in $\text{Na}_{2.26}\text{Fe}_{1.87}(\text{SO}_4)_3$. The GCD curves for both NFS-H and NFS-L (Figure S17a) show two slanted plateau-like features (with an abrupt change at ~ 4.0 V). A tentative explanation of such an anomaly with appropriate citations should be provided.

8. The authors should discuss the capacity fading mechanisms with charge/discharge cycles in the cathode material and the role of increased concentrations of $\text{Na}_6\text{Fe}(\text{SO}_4)_4$ in preventing it.

(see attached document for convenience)

Reviewer #2 (Remarks to the Author):

Article "Bridging multiscale interfaces unlocks ion-conducting pathways for stable high-voltage sodium storage cathode" by Jiyu Zhang et al. reports synthesis of interface engineered $\text{Na}_{2.26}\text{Fe}_{1.87}(\text{SO}_4)_3/\text{Na}_6\text{Fe}(\text{SO}_4)_4$ heterostructure cathode material for Na-ion batteries, its bulk and surface structure, as well as promising performance in an Na-ion full cell with FeS as an anode, delivering good energy density and reversibility.

Previous works often documented precipitation of $\text{Na}_6\text{Fe}(\text{SO}_4)_4$ as a secondary phase in synthesis of $\text{Na}_{2+x}\text{Fe}_{2-x/2}(\text{SO}_4)_3$ materials, however, this work claims synergistic effects arising from occurrence of the two-phase system due to optimized synthesis method. Fast ion conducting properties of $\text{Na}_6\text{Fe}(\text{SO}_4)_4$ phase are claimed based on computer modeling.

I regard this work as novel and insightful, so I recommend to accept for publication Nature Communications after the following comments are appropriately addressed:

1. Lines 138-145. Fast ionic conductivity of $\text{Na}_6\text{Fe}(\text{SO}_4)_4$ phase should be supported experimentally by direct measurement of ionic conductivity of bulk $\text{Na}_6\text{Fe}(\text{SO}_4)_4$ sample, as computer simulations can be imprecise.
2. Lines 138-145. Energy barriers for $\text{Na}_{2.26}\text{Fe}_{1.87}(\text{SO}_4)_3$ phase (3.33 eV) seem to be overestimated and differ significantly from previously published values (0.63 eV) by L.L. Wong Phys.Chem.Chem.Phys. 2015, 17, 9186. Computer model should be corrected, differences should be acknowledged and adequately explained.
3. Lines 177-183, Figure 2f. Position of the diffraction peaks related to the $\text{Na}_6\text{Fe}(\text{SO}_4)_4$ phase marked with a dashed line between 26 and 27 degrees are highly speculative, and as such this part should be omitted from the article. Recent work (J. Mater. Chem. A, 2019, 7, 13197) revealed very low capacity for the $\text{Na}_6\text{Fe}(\text{SO}_4)_4$ phase <15 mAh/g.

Response to Reviewer 1:

The manuscript by Zhang et al. reports on the role of a $\text{Na}_6\text{Fe}(\text{SO}_4)_4$ minority phase in enhancing the electrochemical performance of $\text{Na}_{2.26}\text{Fe}_{1.87}(\text{SO}_4)_3$ cathode material for Na-ion batteries. The manuscript also discussed the impact of the exposure-preferred (11-2) plane of $\text{Na}_{2.26}\text{Fe}_{1.87}(\text{SO}_4)_3$ particles in establishing a stable and high ion conducting SEI layer, which helped them achieve a high capacity of 101.3 mAh g^{-1} with an exceptional rate performance (73.5 mAh g^{-1} at 10 C) and cyclic stability ($\sim 65\%$ capacity retention after 5700 cycles). The authors have used multiple theoretical studies as well as structural & spectroscopic techniques to substantiate their findings and ascertain the formations of highly conducting Na^+ migrations channels within the material. In view of the focus on cheaper and earth-abundant Na-ion batteries (NIBs) cathodes, there is a need to explore a broader range of materials that promise excellent rate-performance & cyclability. This is especially important as the poor Na-ions kinetics in the cathode layer are considered one of the significant challenges in the broader adoption of NIBs at the commercial scale. The results present in this manuscript are a decent addition to the research on NIB cathodes. However, the following points must be addressed in the revised manuscript before being accepted for publication in Nature Communications.

Q1. (a) The authors have used theoretical calculations to substantiate the role of $\text{Na}_6\text{Fe}(\text{SO}_4)_4$ in forming highly conducting Na-ion pathways with the material. From an experimental standpoint, it is highly recommended that the authors substantiate this result through experimental means (EIS, chronoamperometry, etc.) on the as-prepared $\text{Na}_6\text{Fe}(\text{SO}_4)_4$ material to confirm its superionic conducting nature. It would be helpful to the reader and improve the quality of this manuscript if the difference in ionic conductivities of the main $\text{Na}_{2.26}\text{Fe}_{1.87}(\text{SO}_4)_3$ phase and minor $\text{Na}_6\text{Fe}(\text{SO}_4)_4$ phase is quantified. **(b)** In lines 123–127, the authors mention the synthesis of 4.52 wt% $\text{Na}_6\text{Fe}(\text{SO}_4)_4$ phase and the metastable nature of $\text{Na}_{2.26}\text{Fe}_{1.87}(\text{SO}_4)_3$ and $\text{Na}_6\text{Fe}(\text{SO}_4)_4$ phases. Is this valid only for the co-precipitation method or in general? **(c)** Nonetheless, the electrical properties of NFS-H and NFS-L can be compared. Interestingly, DCIR for both cells is quite similar (Fig. 5d) for the first 50-60 cycles!

Our response: We appreciate the reviewer's professional suggestion. We will answer the above questions in three small parts for clear.

(a) According to your suggestion, we prepared the material with $\text{Na}_6\text{Fe}(\text{SO}_4)_4$ as main phase (PDF 29-1218) and conducted its physical and electrochemical characterizations. The electrode's preparation of as-synthesized $\text{Na}_6\text{Fe}(\text{SO}_4)_4$ is same as that of NFS-H electrode. Detailed experimental synthesis steps are as follows: The molar ratio (3:1) of anhydrous NaSO_4 , FeSO_4 and 10 wt.% carbon black materials (Super P Li) are put into the tank in an environment filled with pure Ar gas (99.99%), and then ball milling is carried out. This process lasts for 6 h, and the speed of ball mill is 500 rpm. The obtained material is then calcined in a tubular furnace at $350 \text{ }^\circ\text{C}$ for 12 h with a heating rate of $1 \text{ }^\circ\text{C}/\text{min}$ in Ar gas. Finally, the $\text{Na}_6\text{Fe}(\text{SO}_4)_4$ material is obtained. XRD pattern demonstrates the successful synthesis of $\text{Na}_6\text{Fe}(\text{SO}_4)_4$ phase (**Figure R1a**). Furthermore, the analysis shows that trace amount of NaSO_4 and Na_5FeO_4 phases are also included as an impurity, which might originate from the inexhaustive solid reaction and high-temperature oxidation during synthesis process.

The electrochemical performance of as-synthesized $\text{Na}_6\text{Fe}(\text{SO}_4)_4$ is also studied in CR2025 coin-type cells. The galvanostatic charge/discharge curves (**Figure R1b**) shows a poor sodium storage ability with a low specific capacity of 10 mAh g^{-1} at 12 mA g^{-1} , which is similar to the reported performance

by Jiang et al. (*J. Mater. Chem. A*, **2019**, 7, 13197). Nyquist plots were performed for the prepared $\text{Na}_6\text{Fe}(\text{SO}_4)_4$ electrode after initial cycle with the amplitude of the AC voltage set at 5 mV from 100 kHz to 0.01 Hz (Autolab RRDE/RDE-2). In the fitted equivalent circuit (**Figure R1c**), R1 represents the ohmic resistance in the cell system. Meanwhile, R2 and CPE1 (constant phase angle element) correspond to the resistance and capacitance of interfacial CEI. In addition, R3, CPE2 and W1 represent the faradaic impedance of the reaction, the interphase double layer capacitance and the diffusion impedance, respectively. Based on the Fick's laws (equation R1), the Na^+ diffusion coefficient inside the as-synthesized $\text{Na}_6\text{Fe}(\text{SO}_4)_4$ electrode was calculated to be $10^{-9.59} \text{ cm}^2 \text{ s}^{-1}$.

$$D = \frac{R^2 T^2}{2A^2 n^4 F^4 C^2 \sigma^2} \quad (\text{Equation R1})$$

R is the gas constant, T represents the temperature, A is the area of the electrode, F is the Faraday constant, n is the number of electrons per molecule in the charge-discharge reaction and σ is the slope of the $Z''-\omega^{-1/2}$ curve (**Figure R1d**).

The galvanostatic intermittent titration technique (GITT) of as-synthesized $\text{Na}_6\text{Fe}(\text{SO}_4)_4$ was carried out in a coin cell after 2 cycles to reach the thermal equilibrium state (**Figure R1e**). Based on equation 1 in Methods, the diffusion coefficients of Na^+ ions at different voltage states were calculated and displayed in **Figure R1f**. The diffusion coefficients of $\text{Na}_6\text{Fe}(\text{SO}_4)_4$ electrode range from $10^{-10.54}$ to $10^{-7.21} \text{ cm}^2 \text{ s}^{-1}$ within the voltage window of 2.0–4.5 V, which shows obvious superiority to alluaudite-type $\text{Na}_{2+2x}\text{Fe}_{2-x}(\text{SO}_4)_3$ (Supplementary table 6). For instance, Chou's group reported that the alluaudite-type $\text{Na}_2\text{Fe}_2(\text{SO}_4)_3$ materials without visible $\text{Na}_6\text{Fe}(\text{SO}_4)_3$ impurity featured a diffusion coefficients range from 10^{-12} to $10^{-10.8} \text{ cm}^2 \text{ s}^{-1}$ (*Adv. Energy Mater.*, **2018**, 8, 1800944).

Thus, a superior Na^+ conductivity can be determined in $\text{Na}_6\text{Fe}(\text{SO}_4)_4$ phase, which supports our theoretical calculation and electrochemical results.

Figure R1. The characterizations of as-prepared $\text{Na}_6\text{Fe}(\text{SO}_4)_4$ material. (a) XRD pattern. (b) Charge-discharge curves at 12 mA g^{-1} . (c) EIS plot of prepared $\text{Na}_6\text{Fe}(\text{SO}_4)_4$ electrode after initial cycle (line: fitted data; dot: pristine data). (d) Fitted $Z'-\omega^{-1/2}$ curve. (e) GITT curves. (f) Calculated Na^+ diffusion coefficients.

(b) In fact, the $\text{Na}_6\text{Fe}(\text{SO}_4)_4$ phase is a common concomitant impurity during the synthesis of $\text{Na}_{2+2x}\text{Fe}_{2-x}(\text{SO}_4)_3$. The formation of $\text{Na}_6\text{Fe}(\text{SO}_4)_4$ phase during the synthesis of $\text{Na}_{2+2x}\text{Fe}_{2-x}(\text{SO}_4)_3$ phases might occur in general synthesis methods. The reasons were discussed from the theoretical and experimental results.

Firstly, the formation of $\text{Na}_6\text{Fe}(\text{SO}_4)_4$ phase is largely dependent on thermodynamic equilibrium of phases. In 2019, Teeraphat Watcharatharapong et al. adopted the first-principles calculations combined with a random swapping method to investigate phase stability diagrams and defect formations in $\text{Na}_{2+2x}\text{Fe}_{2-x}(\text{SO}_4)_3$ material systems based on thermodynamic criteria (*ACS Appl. Mater. Interfaces* **2019**, 11, 32856–32868). They listed the reaction entropy formulas of various phases based on chemical potential formalism, which were presented as following:

$$(2+2x) \mu_{\text{Na}} + (2-x) \mu_{\text{Fe}} + 3 \mu_{\text{S}} + 12 \mu_{\text{O}} = \Delta H^{\ddagger}[\text{Na}_{2+2x}\text{Fe}_{2-x}(\text{SO}_4)_3] \quad (\text{Equation R2})$$

$$\mu_{\text{Fe}} + \mu_{\text{S}} + 4 \mu_{\text{O}} < \Delta H^{\ddagger}[\text{FeSO}_4] \quad (\text{Equation R3})$$

$$6 \mu_{\text{Na}} + \mu_{\text{Fe}} + 4 \mu_{\text{S}} + 16 \mu_{\text{O}} < \Delta H^{\ddagger}[\text{Na}_6\text{Fe}(\text{SO}_4)_4] \quad (\text{Equation R4})$$

The calculation result indicates the solution to the equation R2 is not available under the constraints of equation R3 and R4. It suggests that the thermodynamic boundary conditions for $\text{Na}_{2+2x}\text{Fe}_{2-x}(\text{SO}_4)_3$ with $x = 0, 0.25, \text{ and } 0.5$ cannot be solved (equation R2) if the constrains for the competing phases $\text{Na}_6\text{Fe}(\text{SO}_4)_4$ and FeSO_4 are both taken into consideration (equation R3 and R4). Therefore, $\text{Na}_{2+2x}\text{Fe}_{2-x}(\text{SO}_4)_3$ should crystallize with an inevitable formation of either one of these secondary phases (or impurities).

Secondly, the experimental results of formed $\text{Na}_6\text{Fe}(\text{SO}_4)_4$ phase during the synthesis of $\text{Na}_{2+2x}\text{Fe}_{2-x}(\text{SO}_4)_3$ and synthesis methods were summarized from extensive reported literature. Afterwards, we identified two formation mechanisms of $\text{Na}_6\text{Fe}(\text{SO}_4)_4$ phase. (1) The solid-reaction conversation of FeSO_4 and Na_2SO_4 . According to the study by Atsuo Yamada et al., in the Na_2SO_4 – FeSO_4 binary system, solid-state reactions at a moderate temperature of 623 K produce two stable phases: vanthoffite-structured $\text{Na}_6\text{Fe}(\text{SO}_4)_4$ and alluaudite-type $\text{Na}_{2+2x}\text{Fe}_{2-x}(\text{SO}_4)_3$ with a certain non-stoichiometry (*ChemElectroChem*, **2015**, 2, 1019–1023). Their phase relationship is presented in **Scheme R1**. This formation mechanism has been mentioned in the reported $\text{Na}_2\text{Fe}(\text{SO}_4)_2$ (90.5%)/ $\text{Na}_6\text{Fe}(\text{SO}_4)_4$ (9.5%) (*J. Mater. Chem. A*, **2019**, 7, 13197–13204), $\text{Na}_{2+2x}\text{Fe}_{2-x}(\text{SO}_4)_3$ / $\text{Na}_6\text{Fe}(\text{SO}_4)_4$ (*ChemElectroChem*, **2015**, 2, 1019–1023). Their synthesis methods are mainly solid-phase ball milling and calcining process. (2) The decomposition of $\text{Na}_2\text{Fe}(\text{SO}_4)_2 \cdot 4\text{H}_2\text{O}$ precursor derived from equal molar composition of $\text{FeSO}_4 \cdot 7\text{H}_2\text{O}$ and Na_2SO_4 , which decomposes into the $\text{Na}_{2+2x}\text{Fe}_{2-x}(\text{SO}_4)_3$ phase and minor $\text{Na}_6\text{Fe}(\text{SO}_4)_4$ phase at a high temperature. The reaction equation can be expressed in equation R5 and R6. This formation mechanism has been mentioned in the reported $\text{Na}_{2.27}\text{Fe}_{1.86}(\text{SO}_4)_3$ (97.01%)/ $\text{Na}_6\text{Fe}(\text{SO}_4)_4$ (1.01%) (*J. Solid State Electr.*, **2022**, 26, 1941–1950), $\text{Na}_{2.4}\text{Fe}_{1.8}(\text{SO}_4)_3$ / $\text{Na}_6\text{Fe}(\text{SO}_4)_4$ (*J. Energy Chem.*, **2021**, 54, 564–570), $\text{Na}_2\text{Fe}(\text{SO}_4)_2$ (95.6%)/ $\text{Na}_6\text{Fe}(\text{SO}_4)_4$ (4.4%) (*J. Energy Chem.*, **2021**, 54, 564–570), $\text{Na}_{2.7}\text{Fe}_{1.65}(\text{SO}_4)_3$ / $\text{Na}_6\text{Fe}(\text{SO}_4)_4$ (*Electrochim. Acta*, **2019**, 296, 345–354), $\text{Na}_{2.9}\text{Fe}_{1.55}(\text{SO}_4)_3$ (96.4%)/ $\text{Na}_6\text{Fe}(\text{SO}_4)_4$ (3.6%) (*J. Mater. Chem. A*, **2016**, 4, 1624–1631), and so on. Their synthesis methods include co-precipitation and calcining process, spray drying and calcining process, freeze drying and calcining process. Whatever the reaction mechanism is, the existence of $\text{Na}_6\text{Fe}(\text{SO}_4)_4$ phase could be found.

Scheme R1. Schematic phase relationship in the $\text{Na}_2\text{SO}_4\text{-FeSO}_4$ system based on present knowledge. (*ChemElectroChem*, 2015, 2, 1019 - 1023)

(c) According to your suggestion, we further analyzed the electrochemical difference between NFS-H and NFS-L electrodes.

In galvanostatic charge-discharge curves at 60 mA g^{-1} (**Figure R2a**), NFS-H shows a reversible capacity of 85.5 mAh g^{-1} and a high mid-value voltage (3.65 V) at the initial cycle. While, NFS-L features a capacity of 79.8 mAh g^{-1} and a lower mid-value voltage (3.61 V). Benefited from the good ionic-conducting of $\text{Na}_6\text{Fe}(\text{SO}_4)_4$, NFS-H shows a lower potential difference of 1.221 V than that of NFS-L (1.809 V) at 95% discharged capacity (SOC%=5%), which indicates weaker electrochemical polarization and improved kinetics of NFS-H with richer $\text{Na}_6\text{Fe}(\text{SO}_4)_4$ phase. After 50 cycles, NFS-H shows a capacity of 82.3 mAh g^{-1} with a potential difference of 1.240 V, corresponding to a capacity retention of 96.3% (**Figure R2b**). While NFS-L shows a capacity of 76.2 mAh g^{-1} with a potential difference of 1.818 V and a capacity retention of 95.5%. Although their capacity retention is similar, a smaller potential difference is present in NFS-H. The discharged direct-current (DC) resistance of NFS-H at 50th cycle (345 ohm) is also smaller than that of NFS-L (373 ohm) (**Figure R2e**). **However, this difference is largely shielded by exaggerated size scale in original manuscript.**

We also conducted EIS plots on NFS-H and NFS-L electrodes during the initial 50 cycles (**Figure R2f - h**). For the NFS-L, the diffusion impedance (W1) is greatly larger than interfacial resistance (R2) in value, indicating the ions diffusion inside the bulk of electrodes contributes to the major impedance of batteries. However, the case does not occur in NFS-H. Moreover, the result also shows increased R2 and W1. Specifically, the R2 increases 31.3% and 55.1% during the initial 50 cycles for NFS-H and NFS-L, respectively, which mainly originates from the decomposition of electrolytes on cathode surface. And the W1 increases 34.6% and 41.9% during the initial 50 cycles for NFS-H and NFS-L, respectively. The lower increase of diffusion impedance in NFS-H implies better maintenance on reaction reversibility and kinetics within the bulk of cathodes due to introduced ionic-conducting $\text{Na}_6\text{Fe}(\text{SO}_4)_4$ phase.

Relatively short cycling (50 cycles) is incapable of severely damaging the bulk and interface structure of electrodes, enabling similar electrochemical polarization and capacity retention in NFS-H and NFS-L electrodes. However, during a longer cycle, the interface resistance caused by electrolyte's decomposition and the diffusion resistance within the bulk accumulate continuously, resulting in continuously increased potential difference and decreased voltage output of the electrodes. For instance, after 600 cycling, NFS-L shows a capacity retention of 73.5% (**Figure R2c**). And its electrochemical polarization obviously increases, showing a large potential difference of 1.992 V. After 1200 cycles, the specific capacity further decreases to 53.2% of the original capacity (**Figure R2d**). The potential difference of batteries further increases to 2.203 V, resulting the distortion of charge-discharge curves and the disappearance of electrochemical characters in the set voltage range. Differently, for NFS-H cathode, the charge-discharge platform largely maintains the original characteristics during long cycles, including plateau-like feature and an abrupt change at 3.0 V. Moreover, lower increase of potential difference and higher capacity retention are found (1.401 V, 88.1%, after 600 cycles; 1.603 V, 80.5%,

after 1200 cycles). Thus, it is reasonable to infer that, benefited from the rich introduction of ionic-conducting $\text{Na}_6\text{Fe}(\text{SO}_4)_4$ phase, sufficient ion diffusion channels could be effectively built in electrodes. It greatly improves the ion conductivity and electrochemical polarization of NFS-H, and facilitates the good maintenance of reversible (de)insertion reaction and kinetics during long-term cycles, ensuring good reaction reversibility and capacity maintenance of electrodes.

Figure R2. The electrochemical performance of NFS-H and NFS-L at different cycles at 60 mA g⁻¹. The charge-discharge curves of (a) 1st; (b) 50th; (c) 600th; (d) 1200th cycles. The vertical lines indicate the overpotentials of batteries at 95% discharged capacity (SOC%=5%). (e) DC internal resistance. EIS plots after (f) initial cycle, (g) 20 cycles and (h) 50 cycles (line: fitted data; dot: pristine data).

On page 5 of revised manuscript, we added the discussion with “Furthermore, the diffusion coefficients range of as-prepared $\text{Na}_6\text{Fe}(\text{SO}_4)_4$ phase is determined by $10^{-10.543}$ to $10^{-7.213}$ cm² s⁻¹ (Supplementary Figure S9), which shows obvious superiority to pure alluaudite-type $\text{Na}_{2+2x}\text{Fe}_{2-x}(\text{SO}_4)_3$.^{[25]”}.

On page 10 of revised manuscript, we have added the discussion with “NFS-H cathode delivers an exceptional cycle stability, with a capacity retention of 80.69% at 60 mA g⁻¹ after 1300 cycles (127 days, Fig. 5d). Furthermore, the capacity fading mechanisms were discussed based on the coulombic efficiency, EIS plots and the direct-current (DC) resistances. Firstly, NFS-L displays lower initial coulombic efficiency (72.2%) and average coulombic efficiency (98.66%) than those (82.3% and 99.59%) of NFS-H. Meanwhile, the interfacial resistance increases by 31.3% and 55.1% during the initial 50 cycles for NFS-H and NFS-L, respectively (Supplementary Fig. 22). In addition, at 1200th cycle, the discharged

direct-current (DC) resistances of NFS-H and NFS-L show apparent increases by 154.6% and 552.8%. Therefore, the reason of capacity retention is largely ascribed to continuous interfacial side reactions and the thickening of CEI (Supplementary Fig. 18). On the other hand, the insertion/extraction of Na^+ leads to changes in the molar volume of $\text{Na}_{2.26}\text{Fe}_{1.87}(\text{SO}_4)_3$ and $\text{Na}_6\text{Fe}(\text{SO}_4)_4$ phases, which may induce mechanical stress and strain to their crystals. Thus, accidental disconnection between two phases is possible in local regions, which might damage the ion-conduction networks inside the electrodes and boost the diffusion impedance. Therefore, the introduced ionic-conducting $\text{Na}_6\text{Fe}(\text{SO}_4)_4$ phase, which built sufficient ion diffusion channels in NFS-H, could facilitate its good maintenance of reversible (de)insertion reaction and capacity.”

On page 12 of revised manuscript, we added the preparation process of $\text{Na}_6\text{Fe}(\text{SO}_4)_4$ material as “For the preparation of $\text{Na}_6\text{Fe}(\text{SO}_4)_4$ material, the molar ratio (3:1) of anhydrous NaSO_4 , FeSO_4 and 10 wt.% carbon black materials (Super P Li) are put into the tank in an environment filled with pure Ar gas (99.99%), and then ball milling is carried out. This process lasts for 6 h, and the speed of ball mill is set to 500 rpm/min. The obtained material is then calcined in a tubular furnace at 350 °C for 12 h with a heating rate of 1 °C/min in Ar gas. Finally, the $\text{Na}_6\text{Fe}(\text{SO}_4)_4$ material is obtained.”

On page 10 of revised Supplementary information, we added the Figure R1 as **Supplementary Fig. 9**.

On page 23 of revised Supplementary information, we added the Figure R2 as **Supplementary Fig. 22**.

On page 31 of revised Supplementary information, we added the discussion about Supplementary Fig. 9 as **Supplementary Note 2**.

On page 35 of revised Supplementary information, we added the discussion about Supplementary Fig. 22 as **Supplementary Note 6**.

Q2. XRD Analysis

a. Fig. 1(b) & Fig. S2(c) shows the XRD profiles of NFS-H with low signal-to-noise ratios. As this data is instrumental in identifying crystallographic parameters and phase fractions, the authors should obtain better-quality XRD/neutron/Synchrotron XRD data to analyze the material.

b. Also, include the error in the estimated% phase fractions.

c. Fig. 2(f) again shows poor quality data of ex-situ XRD data which makes it difficult to judge the phase formations/structural evolutions discussed by the authors.

d. For the reader's benefit, it is recommended to include details about the experimental setup used for in-situ XRD studies (cell configuration, operando/in-situ conditions, etc.)

Our response: We appreciate the reviewer's suggestions.

(a) According to the reviewer's suggestions, we supplemented the Synchrotron high-pressure powder XRD pattern of NFS-H, which was carried out at the BL14B1 beamline of the Shanghai Synchrotron Radiation Facility (SSRF) using X-rays with a wavelength of 0.6887 Å.

XRD pattern of NFS-H and its Rietveld refinement are shown in **Figure R3**. The result fits into the C2/c space group with the lattice parameters $a = 12.7221$ Å, $b = 12.8437$ Å, $c = 6.5589$ Å, $\beta = 115.555^\circ$ and $V = 966.879$ Å³, and it was found to be isostructural to alluaudite-type $\text{Na}_{2.26}\text{Fe}_{1.87}(\text{SO}_4)_3$. Furthermore, trace amount of $\text{Na}_6\text{Fe}(\text{SO}_4)_4$ (9.33 wt.%) is also included with the lattice parameters $a = 9.702387$ Å, $b = 9.271386$ Å, $c = 8.257282$ Å, $\beta = 113.387^\circ$ and $V = 681.755$ Å³. Detailed crystallographic parameters are presented in Table R1 and Table R2. Furthermore, the atomic ratio of Na:Fe (1.42:1) determined by EDS spectra (Supplementary Fig. 3) is close to the value (1.47:1) calculated from refined XRD result, supporting the analysis of co-existed $\text{Na}_{2.26}\text{Fe}_{1.87}(\text{SO}_4)_3$ and

Na₆Fe(SO₄)₄ phases in NFS-H.

Figure R3. Synchrotron high-pressure powder X-ray diffraction riveted refinement of NFS-H.

Table R1. Atomic parameters of Na_{2.26}Fe_{1.87}(SO₄)₃ refined from Synchrotron high-pressure XRD riveted refinement of NFS-H. Space group: C2/c, *a*=12.7221 Å, *b*=12.8437 Å, *c*=6.5588 Å, *V*=966.879 Å³.

Atom	x	y	z	Occ.	Iso.
Na1	0.500000	0.731537	0.750000	1.0	0.01478
Na2	0.000000	0.000000	0.000000	0.7430	0.03475
Na3	0.500000	0.982422	0.250000	0.5680	0.00460
Fe	0.730724	0.157770	0.147689	0.9359	0.00409
S1	0.000000	0.778764	0.750000	1.0	-0.00509
S2	0.760692	0.599142	0.866545	1.0	-0.00730
O1	0.070117	0.847020	0.710371	1.0	0.02388
O2	0.447019	0.209871	0.545086	1.0	-0.01689
O3	0.766531	0.665477	0.691271	1.0	-0.00203
O4	0.330900	0.998025	0.394337	1.0	0.02520
O5	0.362917	0.583915	0.680470	1.0	0.01652
O6	0.328971	0.150410	0.076752	1.0	0.01725

Table R2. Atomic parameters of Na₆Fe(SO₄)₄ refined from Synchrotron high-pressure powder XRD riveted refinement of NFS-H. Space group: P 21/c, *a*=9.7018 Å, *b*=9.2715 Å, *c*=8.2570 Å, *V*=681.755 Å³.

Atom	x	y	z	Occ.	Iso.
Na1	0.266766	0.007476	0.475326	1.0	-0.04763
Na2	0.437112	0.056891	0.081912	1.0	-0.04681
Na3	0.865898	0.118326	0.353204	1.0	-0.06723
Fe	0.000000	0.000000	0.000000	1.0	-0.05444
S1	0.140301	0.285214	0.252767	1.0	-0.01857
S2	0.644954	0.397857	0.306339	1.0	-0.04014
O1	0.105016	0.294197	0.184144	1.0	-0.08120

O2	0.129494	0.410082	0.086854	1.0	-0.09000
O3	0.143515	0.336069	0.265634	1.0	-0.09000
O4	0.188109	0.264697	0.246820	1.0	-0.09000
O5	0.722423	0.222255	0.312435	1.0	-0.07228
O6	0.606748	0.500504	0.466667	1.0	-0.09000
O7	0.796952	0.432058	0.355587	1.0	-0.09000
O8	0.647600	0.330134	0.364212	1.0	-0.09000

(b) Based on the refined result, the phase/element fraction sum (shift/error)**2 is 3.80.

(c) Sorry for our unclear description. The previous spectra with poor quality in Figure 2f were extracted from the *in-situ* XRD patterns at various voltages. Due to the shielding effect of Be window, the peak signals around 26° show poor quality and strong background noise, which might provide inaccurate information of peak evolution. Accordingly, we further supplemented relevant *ex-situ* XRD patterns of NFS-H at various voltages.

Related experimental details about *ex-situ* XRD patterns are presented as follows: the working electrode was fabricated by mixing the active material, super P and polyvinylidene fluoride with a weight ratio of 80:10:10 in N-methyl-2-pyrrolidone to form a slurry. The obtained slurry was coated on ultra-thin aluminum foil (6 μm, Qiandingli electronic technology company) and dried overnight at 120 °C in vacuum with a loading about 6.0–7.0 mg cm⁻². After the specific cycle, disassemble the coin cells and use propylene carbonate solvent to wash the electrodes three times. The processes of battery disassembly, electrode's washing and drying are carried out in an argon-filled glove box (O₂ and H₂O < 0.1 ppm). *Ex-situ* XRD pattern was performed in X-ray powder diffraction (XRD, BRUKER D8 ADVANCE A25) with Cu Kα radiation (λ=1.54056 Å) at a scan rate of 1 °/min.

The obtained *ex-situ* XRD patterns are presented in **Figure R4**. Compared with the patterns extracted from the *in-situ* XRD patterns at various voltages, the newly supplemental *ex-situ* XRD patterns at various voltages display stronger peak intensity and lower background noise, which enhances the credibility of our discussion. During charged process of Na₆Fe(SO₄)₄, an visible shift around 3.5 V to high degrees marks the emergence of the desodiated Na_{6-x}Fe(SO₄)₄ phase. During the subsequent electrochemical process, it kept a gentle trend of peak evolution and finally returned to the initial peak position after the initial cycle. For Na_{2.26}Fe_{1.87}(SO₄)₃ phase, its peak attributed to the (13-1) plane shows a weak shift to high angels during the whole charge process, and then backs to the original positions upon the completion the full sodiation of Na_{2.26-x}Fe_{1.87}(SO₄)₃. The bend of peak evolution is later than that of Na₆Fe(SO₄)₄ phase, around 4.2 V. Importantly, at high voltages, the peak intensity becomes significantly weaker, which may be attributed to the severe distortion in the crystal structure due to continuous deintercalation of Na⁺ ions. Similar phenomenon is also reported in Na₂Fe₂(SO₄)₃@C@GO (*Adv. Energy Mater.*, **2018**, 8, 1800944) and Na_{2.4}Fe_{1.8}(SO₄)₃ (*J. Energy Chem.*, **2021**, 54, 564–570).

We have replaced the *ex-situ* XRD data with newly supplemental *ex-situ* XRD patterns at various voltages Figure 2g.

Figure R4. Structure evolution of NFS-H. (a) Charge-discharge curve. (b) Newly supplemental *ex-situ* XRD patterns at various voltages. (c) Previous selected *in-situ* XRD patterns at various voltages. (d) *In-situ* XRD patterns. The asterisk represents the $\text{Na}_6\text{Fe}(\text{SO}_4)_4$ material.

(d) We have supplemented related experimental detail about *in-situ* XRD in Materials characterization as follows: *In-situ* XRD pattern was performed in X-ray powder diffraction (XRD, PANalytical Empyrean) with Cu $K\alpha$ radiation ($\lambda=1.54056 \text{ \AA}$) and voltage of 45 kV using an *in-situ* electrochemical device (Beijing Zhongyan Huanke Technology Co., Ltd). The working electrode was fabricated by mixing the active material, super P and polyvinylidene fluoride with a weight ratio of 80:10:10 in N-methyl-2-pyrrolidone to form a slurry. The obtained slurry was coated on ultra-thin aluminum foil ($6 \mu\text{m}$, Qiandingli electronic technology company) and dried overnight at $120 \text{ }^\circ\text{C}$ in vacuum with a loading about $6.0\text{--}7.0 \text{ mg cm}^{-2}$. Other assembled parameters are the same as those of coin cells.

On page 4 of revised manuscript, we have revised the related discussion with “All the diffraction peaks of Synchrotron high-pressure powder X-ray diffraction pattern of NFS-H can be assigned to alluaudite-type $\text{Na}_{2.26}\text{Fe}_{1.87}(\text{SO}_4)_3$ and vanthoffite-type $\text{Na}_6\text{Fe}(\text{SO}_4)_4$ crystals, in which the sharp diffraction peaks indicate high crystallinity (Fig. 1b). Notably, compared with $\text{Na}_{2+2x}\text{Fe}_{2-x}(\text{SO}_4)_3$ that are converted directly from the composite of Na_2SO_4 and FeSO_4 reported in literature,^[25,26] the intermediate-derived $\text{Na}_{2.26}\text{Fe}_{1.87}(\text{SO}_4)_3$ shows a relatively stronger (11-2) peak but weaker (200) peaks. Moreover, Rietveld refinement reveals that the content of $\text{Na}_{2.26}\text{Fe}_{1.87}(\text{SO}_4)_3$ and $\text{Na}_6\text{Fe}(\text{SO}_4)_4$ are deduced to be 90.67 wt.% and 9.33 wt.%, respectively, whose detailed crystallographic parameters are listed in Supplementary table 1–4.”

On page 6 of revised manuscript, we revised the discussion with “*Ex-situ* and *in-situ* XRD patterns further reveal their structure evolutions (Figs. 2f and 2g). During charged process of $\text{Na}_6\text{Fe}(\text{SO}_4)_4$, a subtle shift to high degrees marks the emergence of the desodiated $\text{Na}_{6-x}\text{Fe}(\text{SO}_4)_4$ phase, which continues the shift trend until 3.5 V. After that, characteristic reflections of desodiated $\text{Na}_{2.26-x}\text{Fe}_{1.87}(\text{SO}_4)_3$ appear with the canting of lattice planes, including (13-1), (11-2), (400) and (041) planes. Interestingly, with the beginning of the $\text{Na}_{2.26}\text{Fe}_{1.87}(\text{SO}_4)_3$ desodiation, the diffraction peak of $\text{Na}_{6-x}\text{Fe}(\text{SO}_4)_4$ shifts toward low degree during charged process, indicating partial Na^+ sodiation from the $\text{Na}_{2.26}\text{Fe}_{1.87}(\text{SO}_4)_3$ into $\text{Na}_{6-x}\text{Fe}(\text{SO}_4)_4$. During subsequent electrochemical process, it kept a gentle trend of peak evolution and finally returned to the initial peak position upon the completion the full sodiation of $\text{Na}_{2.26-x}\text{Fe}_{1.87}(\text{SO}_4)_3$.”

On page 13 of revised manuscript, we have supplemented the related experimental detail about *in-situ* and *ex-situ* XRD patterns in Materials characterization with “*In-situ* XRD pattern was performed in an *in-situ* electrochemical device (Beijing Zhongyan Huanke Technology Co., Ltd), and worked in X-ray

powder diffraction (XRD, PANalytical Empyrean) with Cu K α radiation ($\lambda=1.54056 \text{ \AA}$) and voltage of 45 kV. The working electrode was fabricated by mixing the active material, super P and polyvinylidene fluoride (PVDF) with a weight ratio of 80:10:10 in N-methyl-2-pyrrolidone (NMP) to form a slurry. The obtained slurry was coated on ultra-thin aluminum foil (6 μm , Qiandingli electronic technology company) and dried overnight at 120 $^{\circ}\text{C}$ in vacuum with a loading about 6.0–7.0 mg cm^{-2} . Other assembled parameters are the same as those of coin cells. *Ex-situ* XRD pattern was performed in X-ray powder diffraction (XRD, BRUKER D8 ADVANCE A25) with Cu K α radiation ($\lambda=1.54056 \text{ \AA}$) at a scan rate of 1 $^{\circ}/\text{min}$. The electrode was prepared by the same process as above, and was washed by propylene carbonate solvent. The processes of battery disassembly, electrode's washing and drying are carried out in an argon-filled glove box (O_2 and $\text{H}_2\text{O} < 0.1 \text{ ppm}$). ”.

On page 17 of revised manuscript, we have added the Acknowledgement with “The Modern Analysis and Gene Sequencing Center of Zhengzhou University, the Institute of Microstructure and Properties of Advanced Materials at Beijing University of Technology, Shanghai Synchrotron Radiation Facility (SSRF) were appreciated for their support in tests.”

On page 18 of revised manuscript, we have replaced the X-ray diffraction Rietveld refinement of NFS-H with Synchrotron high-pressure powder X-ray diffraction Rietveld refinement in **Fig.1**.

On page 19 of revised manuscript, we have replaced the *ex-situ* XRD patterns with newly supplemental *ex-situ* XRD patterns at various voltages in **Fig. 2**.

On page 3 of revised Supplementary information, we have removed the X-ray diffraction Rietveld refinement pattern of NFS-H into **Supplementary Fig. 2**.

On page 25 of revised Supplementary information, we have added the Table R1 and R2 as the **Supplementary table 1 and 2**, respectively.

Q3. Line 158: The authors have calculated the carbon content in the as-prepared material to be about 12.64 wt% of the total sample. Why was an additional 20% super-P added while preparing the cathode slurry?

Our response: Thanks for your kind consideration. As we all know, conductive carbon is an effective way to construct high-conductivity networks for electron transport of electrode active materials not only as coating layer on the materials but also as additive of slurry preparation. The up-mentioned dual role of carbon can be found in many reported literatures. Some of them are listed as follows: $\text{Na}_2\text{Fe}_2(\text{SO}_4)_3@\text{C}@\text{GO}$ (*Adv. Energy Mater.* **2018**, 8, 1800944), monoclinic $\text{Fe}_2(\text{SO}_4)_3$ (*J. Power Sources* **2019**, 434, 226750), $\text{Na}_2\text{Fe}(\text{SO}_4)_2/\text{C}$ (*J. Mater. Chem. A*, **2019**, 7, 13197), $\text{Na}_2\text{Fe}(\text{SO}_4)_2@\text{rGO}/\text{C}$ (*J. Energy Chem.* **2020**, 50, 387–394), $\text{Na}_{2.4}\text{Fe}_{1.8}(\text{SO}_4)_3@\text{rGO}$ (*J. Energy Chem.* **2021**, 54, 564–570), alluaudite-type $\text{Na}_2\text{Mn}_{3-x}\text{Al}_x(\text{VO}_4)_3$ (*Chem. Mater.* **2022**, 34, 9, 4088–4103).

In this work, poor electron conductivities of polyanion-type iron-based sulfates is a key problem for their applications, resulting in inferior rate performance of batteries. Therefore, for giving full play to the sodium storage potential of active materials, we introduced up-mentioned dual role of carbon. Together, they have built a continuous and highly efficient conductive network in both internal bulk and external boundary of active materials. Their respective functions are described below:

(1) The carbon materials within active materials. It is mainly composed of two kinds of carbon materials, namely, reduced graphene oxide layer and amorphous carbons derived from the calcination of organic materials (citric acid monohydrate and a handful of ascorbic acid). As shown in Supplementary Fig. 10, amorphous carbon is mainly distributed between primary nanoparticles with tight connections,

and the rGO sheets closely cover the macron particles of active materials. They effectively constitute a three-dimensional, continuous carbon conductive network within the active materials.

(2) Additional conductive carbon in electrode. Conductive carbon is an important part of electrodes in batteries, which can transmit electrons and reduce electrochemical polarization. Specifically, it can reduce the contact resistance of electrodes and accelerate the movement of electrons via collecting microcurrent between active materials, active materials and collectors. In addition, the conductive carbon can improve the processing of electrodes and the infiltration of electrolytes. For micro-sized particles of active materials (NFS-H), sufficient conductive carbon can effectively connect the micro-sized particles of active materials, constituting a good conductive network in electrode.

Q4. The total capacitive contribution of NFS-H calculated from the CV curves was > 90% at a scan rate of 0.1 mV/s. The values for peaks 2 and 3 (Figure S18a) are typical for Na-ion batteries. If most of the contribution to specific capacity was from a surface-based phenomenon, what is the relevance of $\text{Na}_6\text{Fe}(\text{SO}_4)_4$ that ostensibly establish high Na-ion conduction pathways in the bulk of the material?

Our response: Thanks for your professional and good question, which will help us to improve the accuracy of our relevant statements. Indeed, the imprecise description of capacitive contribution in the manuscript causes confusion of understanding.

The concept of pseudocapacitor in batteries was proposed by Bruce Dunn et al. (*J. Phys. Chem. C* **2007**, 111, 14925–14931), whose contribution in nature was evidenced by the area normalized capacitance above $100 \mu\text{F}/\text{cm}^2$. Based on the normalization formula for CV kinetics analysis, the electrochemical process in these "embedded pseudocapacitor" materials is not controlled by diffusion. We think that this charge storage contribution is better described by non-diffusion control, instead of capacitor. Therefore, it is more realistic to describe this contribution in terms of **non-diffusion-controlled process** (rapid redox process and electric double layer capacitance). In addition, related view was also reported by Prof. Liu in a review (Definitions of Pseudocapacitive Materials: A Brief Review, *Energy Environ. Mater.* **2019**, 2, 30–37).

In this work, the obtained NFS-H material feature micron-sized (6–8 μm) polyhedra consisting of massively **stacked nanoparticles**, showing high non-diffusion-controlled charge contribution. As the CV curves indicate, non-diffusion processes contribute most of the charge transport in NFS-H, with > 90% at a scan rate of 0.1 mV/s. The reasons might include two points:

- (1) The obtained NFS-H material feature micron-sized polyhedra consisting of massively stacked nanoparticles. The nanoscale size of active material in NFS-H leads to short diffusion lengths for Na ions in electrodes (even negligible ion diffusion, approaching a surface process).
- (2) The introduction of minor ionic-conductive $\text{Na}_6\text{Fe}(\text{SO}_4)_4$ phase improves the ionic kinetics of NFS-H inside particle bulk, greatly accelerating the redox kinetics during electrochemical process, which prevents it from being a major barrier for Na^+ diffusion at full scale in batteries.

On page 34 of revised Supplementary information, we have added related description as **Supplementary Note 5** with "...The reason for high non-diffusion-controlled charge contribution may include two points: (1) The obtained NFS-H material feature micron-sized polyhedra consisting of massively stacked nanoparticles. The nanoscale size of active material in NFS-H leads to short diffusion lengths for Na ions in electrodes (even negligible ion diffusion, approaching a surface process). (2) The introduction of minor ionic-conductive $\text{Na}_6\text{Fe}(\text{SO}_4)_4$ phase improves the ionic kinetics of NFS-H inside particle bulk, greatly accelerating the redox kinetics during electrochemical process, which prevents it from being a major

barrier for Na⁺ diffusion at full scale in batteries.”

Q5. The authors should clarify the mode of existence of the Na₆Fe(SO₄)₄ phase, that is, if this has intergrowth within the Na_{2.26}Fe_{1.87}(SO₄)₃ crystallites (Figure 1a) or if it segregates at the grain boundaries of Na_{2.26}Fe_{1.87}(SO₄)₃ phase (Figure 2g)!

Our response: We thank the reviewer for pointing out this missing information. HAADF-STEM images identify the fine crystal fringes spread inside the NFS-H particle, following the highly crystalline characteristic indicated by XRD pattern (Figure R5a). As observed, the lattice fringes of 0.277 and 0.308 nm match well with (240) and (11-2) planes of Na_{2.26}Fe_{1.87}(SO₄)₃, respectively, and abundant (11-2) planes are observed to be exposed at the periphery of the particles. Furthermore, the heterostructure between Na_{2.26}Fe_{1.87}(SO₄)₃ and Na₆Fe(SO₄)₄ with close contact is visible with obvious grain boundaries, in which the lattice fringes of 0.341 and 0.358 nm are well matched with the (21-2) plane of Na₆Fe(SO₄)₄ and the (13-1) plane of Na_{2.26}Fe_{1.87}(SO₄)₃, respectively (Figure R5b). Corresponding “bright-bright-dark” arrangements resolve two types of bright spot arrangement (Figure R5c).

Based on the HAADF-STEM images of NFS-H, it is clearly observed that the Na₆Fe(SO₄)₄ phase segregates at the grain boundaries of Na_{2.26}Fe_{1.87}(SO₄)₃ phase.

Figure R5. HAADF-STEM images (a and b) of NFS-H materials. (c) Enlarged image of b.

On page 6 of revised manuscript, we have revised the description with “Furthermore, the heterostructure between Na_{2.26}Fe_{1.87}(SO₄)₃ and Na₆Fe(SO₄)₄ with close contact is visible with obvious grain boundaries, in which the lattice fringes of 0.341 and 0.358 nm are well matched with the (21-2) plane of Na₆Fe(SO₄)₄ and the (13-1) plane of Na_{2.26}Fe_{1.87}(SO₄)₃, respectively (Fig. 2b)”

Q6. The SEM image of the “particle,” shown in Fig. 1b inset, seems to be in the form of a polyhedron with 6 dominating faces (rhombic hexahedron). Do all these faces belong to (11-2) family of the monoclinic NFS-H? Interestingly, the particle morphology for NFS-L (Figure S6) sample is drastically different, with no apparent faceting, despite both samples having the same phases.

Our response: Thank you for your good question. The exposed crystal planes of NFS-H were further confirmed via HAADF-STEM images at various regions (Figure R6a – c). The result indicates the rich distribution of (11-2) and (240) crystal planes within NFS-H particles. Except them, other crystal planes like (13-1), (31-3) and (200) are also observed. This is highly consistent with the information shown by XRD results. As we know, the information of crystal planes obtained via XRD includes many particles

from surface to body. Thus, it can be reasonably inferred that in NFS-H materials, not all exposed crystal planes exist in (11-2) structure. But, it is the most exposed one. Furthermore, compared with $\text{Na}_{2+2x}\text{Fe}_x(\text{SO}_4)_3$ reported in literatures (such as, *J. Mater. Chem. A*, **2019**, 7, 14656–14669; *J. Power Sources*, **2020**, 453, 227879; *J. Mater. Chem. A*, **2019**, 7, 13197–13204), NFS-H shows a relatively stronger (11-2) peak (**Figure R6d – g**).

As indicated by the reviewers, the particle morphology of NFS-L is indeed different from that of NFS-H. Specifically, NFS-H shows micron-sized (6–8 μm) polyhedra consisting of massively stacked nanoparticles and rGO sheets attached to the surface. While NFS-L features nano-sized particles with composite rGO sheets, thus, with no apparent faceting. **The morphology difference originates from ethylene glycol (EG) in the solutions.** For NFS-H, the added EG in the solution effectively links the nano-sized precursor ($\text{Na}_2\text{Fe}(\text{SO}_4)_4 \cdot 4\text{H}_2\text{O}$) particles via the interactions between the water groups in precursors and the OH^- bonds at the ends of the molecule itself (*Nano Energy*, **2019**, 56, 502 – 511), which facilitates the self-assembly of nano-sized particles during co-precipitation process and the formation of micron-sized precursor particles. On the other hand, the NFS-H and NFS-L feature the same phase compositions due to **their same formation mechanism (the decomposition derived from the $\text{Na}_2\text{Fe}(\text{SO}_4)_4 \cdot 4\text{H}_2\text{O}$ precursor) and preparation conditions.** Nevertheless, the addition of EG increases the viscosity of solutions, avoiding the single-phase precipitated $\text{FeSO}_4 \cdot 4\text{H}_2\text{O}$ in precursor (as shown in Supplementary Fig. 6). It encourages the formation of $\text{Na}_6\text{Fe}(\text{SO}_4)_4$ phases in the calculated products, enabling a richer ionic-conducting phase in NFS-H.

Figure R6. (a–c) HAADF-STEM images of NFS-H. (d) XRD pattern of NFS-H. (e–g) Referenced XRD

patterns of $\text{Na}_{2+2x}\text{Fe}_{2-x}(\text{SO}_4)_3$ materials in reported literature (*J. Mater. Chem. A*, **2019**, 7, 14656–14669; *J. Power Sources*, **2020**, 453, 227879; *J. Mater. Chem. A*, **2019**, 7, 13197–13204).

On page 6 of revised manuscript, we have added the discussion with “As observed, the lattice fringes of 0.277 and 0.308 nm match well with (240) and (11-2) planes of $\text{Na}_{2.26}\text{Fe}_{1.87}(\text{SO}_4)_3$, respectively, and abundant (11-2) planes are observed to be exposed at the periphery of the particles. Except it, other crystal planes like (240), (13-1) and (200) are also observed (Supplementary Fig. 11).”.

On page 12 of revised Supplementary information, we have added the Figure R6(a–c) as **Supplementary Fig. 11**.

On page 32 of revised Supplementary information, we have added the discussion as **Supplementary Note 3** with “The exposed crystal planes of NFS-H materials were further confirmed via HAADF-STEM images at various regions. The result indicates the rich distribution of (11-2) and (240) crystal planes of NFS-H particle. Except them, other crystal planes like (13-1), (31-3) and (200) are also observed. This is highly consistent with the information shown by XRD results. As we known, the information of crystal planes obtained via XRD include many particles from surface to body. Thus, it can be reasonably inferred that in NFS-H materials, not all exposed crystal planes exist in (11-2) structure. But, it is the most exposed one. Furthermore, compared with $\text{Na}_{2+2x}\text{Fe}_{2-x}(\text{SO}_4)_3$ reported in literatures (such as, *J. Mater. Chem. A*, **2019**, 7, 14656–14669; *J. Power Sources*, **2020**, 453, 227879; *J. Mater. Chem. A*, **2019**, 7, 13197–13204), NFS-H shows a relatively stronger (11-2) peak.”.

Q7. Line 187, Supplementary Information: “In Figure S17a... “. S17a should be changed to S17b. The redox behavior of Fe^{2+} in the minor $\text{Na}_6\text{Fe}(\text{SO}_4)_4$ needs more discussion. The entire capacity is attributed to the de/intercalation accompanied by $\text{Fe}^{2+}/\text{Fe}^{3+}$ transformation in $\text{Na}_{2.26}\text{Fe}_{1.87}(\text{SO}_4)_3$. The GCD curves for both NFS-H and NFS-L (Figure S17a) show two slanted plateau-like features (with an abrupt change at ~4.0 V). A tentative explanation of such an anomaly with appropriate citations should be provided.

Our response: Thanks for your kind consideration and careful examination. According to your suggestions, we have corrected the relevant errors and carried out more discussion and analysis on the electrochemical process of the electrodes.

The redox behavior of Fe^{2+} in the minor $\text{Na}_6\text{Fe}(\text{SO}_4)_4$ could be analyzed from three points: (1) In galvanostatic charge-discharge test at 6 mA g^{-1} , NFS-H shows higher reversible capacity (101.3 mAh g^{-1}), mid-value voltage (3.75 V) and a lower overpotential (1.098 V) than NFS-L (89.7 mAh g^{-1} , 3.66 V, 1.541 V) (**Figure R7a**), indicating the improved reaction kinetics and decreased electrochemical polarization. At high currents, NFS-H shows an obvious gauffer around 3.0 V in discharged curves (**Figure R7b**). According to the report by Jiang et al., it may be attributed to the $\text{Fe}^{2+}/\text{Fe}^{3+}$ redox in $\text{Na}_6\text{Fe}(\text{SO}_4)_4@C$ cathode (*J. Mater. Chem. A*, **2019**, 7, 13197). With the current increases, the electrochemistry belonging to $\text{Na}_6\text{Fe}(\text{SO}_4)_4$ phase shows more and more obvious characteristics, indicating the possibly key role of $\text{Na}_6\text{Fe}(\text{SO}_4)_4$ phase in reaction kinetics at high currents. While NFS-L does not show the gauffer at high currents (**Figure R7c**). In order to show this more clearly, we have gathered the charge-discharge curves $\text{Na}_6\text{Fe}(\text{SO}_4)_4@C$ from the reported literature (**Figure R7d**), in which a discernable gauffer occurs in its discharged curves around 3.0 V. (2) Low sodium storage capacity of $\text{Na}_6\text{Fe}(\text{SO}_4)_4@C$ cathode might indicate poor $\text{Fe}^{2+}/\text{Fe}^{3+}$ redox ability and minor transformation, which is also supported by a subtle shift to high degree of $\text{Na}_6\text{Fe}(\text{SO}_4)_4$ peak around 3.5 V upon charged process in *ex-situ* XRD patterns (**Figure R4**). (3) Peaks located at 2.89/3.05 V and 2.62/2.83 V in CV curves of NFS-H and NFS-

L (**Figure R7e - f**) might be another electrochemical characteristic of $\text{Fe}^{2+}/\text{Fe}^{3+}$ redox in $\text{Na}_6\text{Fe}(\text{SO}_4)_4$ phase (*J. Energy Chem.*, **2020**, 50, 387–394).

Reported $\text{Na}_6\text{Fe}(\text{SO}_4)_4$ phase possesses poor storage sodium activity, whose discharged capacity is only 15 mAh g^{-1} with a working voltage region of $1.5 - 4.2 \text{ V}$ at 4.5 mA g^{-1} (*J. Mater. Chem. A*, **2019**, 7, 13197). Thus, the sodium storage capacity of NFS-H is not entirely attributed to the de/intercalation accompanied by $\text{Fe}^{2+}/\text{Fe}^{3+}$ transformation in $\text{Na}_{2.26}\text{Fe}_{1.87}(\text{SO}_4)_3$, with a small amount contribution from $\text{Fe}^{2+}/\text{Fe}^{3+}$ transformation in $\text{Na}_6\text{Fe}(\text{SO}_4)_4$ phase.

The GCD curves for both NFS-H and NFS-L show two slanted plateau-like features (with an abrupt change at $\sim 4.0 \text{ V}$). The sloping voltage curve over the entire range of Na composition suggests a single-phase homogeneous reaction mechanism involving minimal volume change. According to the result reported by Atsuo Yamada., throughout the whole initial charging process of alluaudite-type $\text{Na}_{2+2x}\text{Fe}_{2-x}(\text{SO}_4)_3$ compounds, Na extraction occurs primarily at Na3 followed by the Na1 and Na2 sites (*Chem. Mater.*, **2016**, 28, 5321–5328). Specially, the broad peak observed at 3.67 V corresponds to Na extraction primarily occurred at the Na3 site with a small migration energy between Na3–Na3 sites. While the bulge at the voltage curve at 3.9 V is attributed to the localized lattice distortion due to the inherent reflection for the Fe migration into vacant Na(1) sites (*Adv. Energy Mater.*, **2018**, 8, 1800944). On the sharp peak in the CV curve at 4.06 V upon the initial charged process, it is associated with Na extraction from Na1 site, which then starts to induce Fe^{3+} migration from the Fe site into the vacant Na1 site (**Figure R7e**).

Figure R7. (a) The charge-discharge curves of NFS-H and NFS-L at 6 mA g^{-1} . The vertical lines indicate the overpotentials of batteries at 95% discharged capacity (SOC%=5%). (b) Rate performance of NFS-H electrode. (c) Rate performance of NFS-L electrode. (d) Reported charge-discharge curves of $\text{Na}_6\text{Fe}(\text{SO}_4)_4/\text{C}$ at 4.5 mA g^{-1} (*J. Mater. Chem. A*, **2019**, 7, 13197). (e) CV curves of NFS-H and NFS-L electrodes. (f) Enlarged CV curves.

On page 9 of revised manuscript, we have added the discussion with “The charge-discharge curves of NFS-H cathode show a reversible capacity of 101.3 mAh g^{-1} and a high mid-value voltage of 3.75 V at initial cycle (Supplementary Fig. 20). In contrast, the NFS-L with less ionic-conducting phase features a bigger electrochemical polarization and a lower capacity (89.7 mAh g^{-1}). It indicates the improved reaction kinetics of ionic-conducting $\text{Na}_6\text{Fe}(\text{SO}_4)_4$ phase, facilitating the decreased electrochemical polarization. Moreover, low sodium storage capacity of $\text{Na}_6\text{Fe}(\text{SO}_4)_4$ cathode might imply poor $\text{Fe}^{2+}/\text{Fe}^{3+}$

redox ability and minor transformation (Supplementary Fig. 9).^[26] Thus, the sodium storage capacity of NFS-H is not entirely attributed to the de/intercalation accompanied by $\text{Fe}^{2+}/\text{Fe}^{3+}$ transformation in $\text{Na}_{2.26}\text{Fe}_{1.87}(\text{SO}_4)_3$, with a small amount contribution from $\text{Na}_6\text{Fe}(\text{SO}_4)_4$ phase. At higher currents, NFS-H shows a discernable gauffer around 3.0 V in charge-discharge curves, which might be an electrochemical characteristic of $\text{Fe}^{2+}/\text{Fe}^{3+}$ redox of $\text{Na}_6\text{Fe}(\text{SO}_4)_4$ phase in NFS-H.^[26] The pairs of weak peaks at 2.89/3.05 V and 2.62/2.83 V in CV curves of NFS-H and NFS-L may be caused by trace $\text{Na}_6\text{Fe}(\text{SO}_4)_4$ phase.^[37] The charge-discharge curves for both NFS-H and NFS-L show two slanted plateau-like features (with an abrupt change at ~ 4.0 V). The sloping voltage curve over the entire range of Na composition suggests the (de)sodiation of Na^+ and a single-phase homogeneous reaction mechanism involving minimal volume change.^[25, 42]

On page 15 - 16 of revised manuscript, we have added two citations in discussion with “[25] M. Chen, D. Cortie, Z. Hu, H. Jin, S. Wang, Q. Gu, W. Hua, E. Wang, W. Lai, L. Chen, S. Chou, X. Wang, S. Dou, *Adv. Energy Mater.* **2018**, 8, 1800944. [42] G. Oyama, O. Pecher, K. J. Griffith, S. I. Nishimura, R. Pigliapochi, C. P. Grey, A. Yamada, *Chem. Mater.* **2016**, 28, 5321.”

On page 21 of revised Supplementary information, we have revised the error in figure legend.

On page 21 of revised Supplementary information, we have added the Figure R7 as **Supplementary Fig. 20**.

On page 33 of revised Supplementary information, we have added related discussion as **Supplementary Note 4**.

Q8. The authors should discuss the capacity fading mechanisms with charge/discharge cycles in the cathode material and the role of increased concentrations of $\text{Na}_6\text{Fe}(\text{SO}_4)_4$ in preventing it.

Our response: Thanks for your kind consideration. According to your suggestion, we further analyzed the charge-discharge curves of cathode materials in different cycles. The results are presented as follows:

Upon the initial cycle, NFS-H shows a similar specific capacity (85.5 mAh g^{-1}) and working voltages (average discharged voltage of 3.65 V) with NFS-L (79.8 mAh g^{-1} and 3.61 V) at 60 mA g^{-1} , but an obviously lower potential difference exists in NFS-H (1.221 V vs. 1.809 V, **Figure R2a**). After 600 cycles, the potential difference of NFS-H and NFS-L increases to 1.401 V and 1.992 V, respectively (**Figure R2c**), which further increases to 1.603 V and 2.203 V after 1200 cycles (**Figure R2d**). Furthermore, we conducted EIS plots on NFS-H and NFS-L during the initial 50 cycles (**Figure R2f-h**), whose result shows continuously increased interfacial resistances (R2) and diffusion impedance (W1). Thus, the capacity fading of cathodes during long-term cycles is mainly from the increased electrochemical polarization. Detailed reasons might include two points:

(1) Interfacial evolution. Upon the initial charging process, the electrolyte is oxidized on cathode surfaces to form a protective CEI layer. And a lower initial coulombic efficiency (72.2%) of NFS-L indicates a more severe decomposition of electrolytes (**Figure R8a**), which might induce the formation of a thicker CEI and bring greater barrier for ion transfer across the interface. In subsequent cycles, side reactions may continuously occur at cathode/electrolyte interface, accompanied by the thickening of CEI and increased interfacial resistance. For example, after 70 cycles at 60 mA g^{-1} , the thickness of CEI formed on NFS-H increases to 12–17 nm from 10–14 nm at 1st cycle (Supplementary Fig. 18). And the interfacial resistance (R2) increases by 31.3% and 55.1% for NFS-H and NFS-L during the initial 50 cycles, respectively.

(2) The fading of heterostructure within bulk. During repeated cycles, the insertion/extraction of

Na^+ leads to changes in the molar volume of $\text{Na}_{2.26}\text{Fe}_{1.87}(\text{SO}_4)_3$ and $\text{Na}_6\text{Fe}(\text{SO}_4)_4$ phases, which may induce mechanical stress and strain to their crystals. Thus, accidental disconnection between two phases is possible in local regions, which might damage the ion-conduction networks and boost the energy barrier of Na^+ migration inside the electrodes. EIS plots also reveal increased diffusion impedance by 34.6% and 41.9% for NFS-H and NFS-L during the initial 50 cycles, respectively. Moreover, due to insufficient ion diffusion channels, sluggish reaction kinetics of NFS-L might maintain from initial cycle and induce continuous reaction irreversibility.

Therefore, during long-time cycle, the interface resistance caused by electrolyte's decomposition and the diffusion resistance within the bulk accumulate continuously, resulting in increased potential difference and decreased capacity output of the electrodes.

Although continued electrochemical fading occurs, remarkable improvement in electrochemical durability is still achieved in NFS-H due to richer ionic-conducting $\text{Na}_6\text{Fe}(\text{SO}_4)_4$ phase. During 1200 cycles at 60 mA g^{-1} , NFS-H achieves a higher average coulombic efficiency of 99.59% than that (98.66%) of NFS-L, which may be attributed to the enhanced reaction reversibility and kinetics (Figure R8b). For instance, compared to the NFS-L, the NFS-H shows a lower potential difference (1.603 V vs. 2.203 V at 1200th cycle) and an increase of diffusion impedance (34.6% vs. 41.9% after 50 cycles), indicating its excellent maintaining of electrochemical polarization. In addition, for the NFS-L after the cycles (1st, 20th and 50th), the diffusion impedance (W1) is greatly larger than interfacial resistance (R2) in value, indicating the ions diffusion inside the bulk of electrodes contributes the major impedance of batteries. However, the case did not occur in NFS-H. Therefore, it is reasonably inferred that, with the increase of introduced ionic-conducting $\text{Na}_6\text{Fe}(\text{SO}_4)_4$ phase, sufficient ion diffusion channels could be effectively built in electrodes. It facilitates the good maintenance of fast reaction kinetics, ensuring good reaction reversibility and capacity maintenance of electrodes.

Figure R8. (a) Initial charge-discharge curves of NFS-H and NFS-L at 6 mA g^{-1} . (b) Coulombic efficiency of NFS-H and NFS-L at 60 mA g^{-1} .

On page 10 of revised manuscript, we have added the discussion with “NFS-H cathode delivers an exceptional cycle stability, with a capacity retention of 80.69% at 60 mA g^{-1} after 1300 cycles (127 days, Fig. 5d). Furthermore, the capacity fading mechanisms were discussed based on the coulombic efficiency, EIS plots and direct-current (DC) resistances. Firstly, NFS-L displays lower initial coulombic efficiency (72.2%) and average coulombic efficiency (98.66%) than NFS-H (82.3% and 99.59%). Meanwhile, the interfacial resistance increases by 31.3% and 55.1% during the initial 50 cycles for NFS-H and NFS-L, respectively (Supplementary Fig. 22). In addition, at 1200th cycle, the discharged DC resistances of NFS-H and NFS-L show apparent increases by 154.6% and 552.8%. Therefore, the reason for capacity fading is largely ascribed to continuous interfacial side reactions and the thickening of CEI (Supplementary Fig.

18). On the other hand, the insertion/extraction of Na^+ leads to changes in the molar volume of $\text{Na}_{2.26}\text{Fe}_{1.87}(\text{SO}_4)_3$ and $\text{Na}_6\text{Fe}(\text{SO}_4)_4$ phases, which may induce mechanical stress and strain to their crystals. Thus, accidental disconnection between two phases is possible in local regions, which might damage the ion-conduction networks inside the electrodes and boost the diffusion impedance. While the introduced ionic-conducting $\text{Na}_6\text{Fe}(\text{SO}_4)_4$ phase, which built sufficient ion diffusion channels in NFS-H, could facilitate its good maintenance of reversible (de)insertion reaction and capacity.”

On page 23 of Supplementary information, we have added the Figure R2 in **Supplementary Fig. 22**.

On page 23 of Supplementary information, we have added the Figure R8 in **Supplementary Fig. 22**.

On page 35 of Supplementary information, we have added the related discussion as **Supplementary Note 6**.

Response to Reviewer 2:

Article “Bridging multiscale interfaces unlocks ion-conducting pathways for stable high-voltage sodium storage cathode” by Jiyu Zhang et al. “reports synthesis of interface engineered $\text{Na}_{2.26}\text{Fe}_{1.87}(\text{SO}_4)_3/\text{Na}_6\text{Fe}(\text{SO}_4)_4$ heterostructure cathode material for Na-ion batteries, its bulk and surface structure, as well as promising performance in an Na-ion full cell with FeS as an anode, delivering good energy density and reversibility.

Previous works often documented precipitation of $\text{Na}_6\text{Fe}(\text{SO}_4)_4$ as a secondary phase in synthesis of $\text{Na}_{2+x}\text{Fe}_{2-x/2}(\text{SO}_4)_3$ materials, however, this work claims synergistic effects arising from occurrence of the two-phase system due to optimized synthesis method. Fast ion conducting properties of $\text{Na}_6\text{Fe}(\text{SO}_4)_4$ phase are claimed based on computer modeling.

I regard this work as novel and insightful, so I recommend to accept for publication Nature Communications after the following comments are appropriately addressed:

Q1. Lines 138-145. Fast ionic conductivity of $\text{Na}_6\text{Fe}(\text{SO}_4)_4$ phase should be supported experimentally by direct measurement of ionic conductivity of bulk $\text{Na}_6\text{Fe}(\text{SO}_4)_4$ sample, as computer simulations can be imprecise.

Our response: Thanks for your consideration. According to your suggestion, we prepared the material with $\text{Na}_6\text{Fe}(\text{SO}_4)_4$ as main phase (PDF 29-1218) and conducted its physical and electrochemical characterizations. The electrode’s preparation of as-synthesized $\text{Na}_6\text{Fe}(\text{SO}_4)_4$ is same as that of NFS-H electrode. Detailed experimental synthesis steps are as follows: The molar ratio (3:1) of anhydrous NaSO_4 , FeSO_4 and 10 wt.% carbon black materials (Super P Li) are put into the tank in an environment filled with pure Ar gas (99.99%), and then ball milling is carried out. This process lasts for 6 h, and the speed of ball mill is 500 rpm. The obtained material is then calcined in a tubular furnace at 350 °C for 12 h with a heating rate of 1 °C/min in Ar gas. Finally, the $\text{Na}_6\text{Fe}(\text{SO}_4)_4$ material is obtained. XRD pattern demonstrates the successful synthesis of $\text{Na}_6\text{Fe}(\text{SO}_4)_4$ phase (**Figure R9a**). Furthermore, the analysis shows that trace amount of NaSO_4 and Na_5FeO_4 phases are also included as an impurity, which might originate from the inexhaustive solid reaction and high-temperature oxidation during synthesis process.

Nyquist plots were performed for the prepared $\text{Na}_6\text{Fe}(\text{SO}_4)_4$ electrode after initial cycle with the amplitude of the AC voltage set at 5 mV from 100 kHz to 0.01 Hz (Autolab RRDE/RDE-2). In the fitted equivalent circuit (**Figure R9b**), R1 represents the ohmic resistance in the cell system. Meanwhile, R2 and CPE1 (constant phase angle element) correspond to the resistance and capacitance of interfacial CEI. In addition, R3, CPE2 and W1 represent the faradaic impedance of the reaction, the interphase double layer capacitance and the diffusion impedance, respectively. Based on the Fick’s laws (equation R1), the Na^+ diffusion coefficient inside the as-synthesized $\text{Na}_6\text{Fe}(\text{SO}_4)_4$ electrode was calculated to be $10^{-9.59} \text{ cm}^2 \text{ s}^{-1}$.

$$D = \frac{R^2 T^2}{2A^2 n^4 F^4 C^2 \sigma^2} \quad (\text{Equation R1})$$

R is the gas constant, T represents the temperature, A is the area of the electrode, F is the Faraday constant, n is the number of electrons per molecule in the charge-discharge reaction and σ is the slope of the $Z''-\omega^{-1/2}$ curve (**Figure R9c**).

The galvanostatic intermittent titration technique (GITT) of as-synthesized $\text{Na}_6\text{Fe}(\text{SO}_4)_4$ was carried out in a coin cell after 2 cycles to reach the thermal equilibrium state (**Figure R9d**). Based on equation

1 in Methods, the diffusion coefficients of Na^+ ions at different voltage states were calculated and displayed in **Figure R9e**. The diffusion coefficients of $\text{Na}_6\text{Fe}(\text{SO}_4)_4$ electrode range from $10^{-10.54}$ to $10^{-7.21}$ $\text{cm}^2 \text{s}^{-1}$ within the voltage window of 2.0–4.5 V, which shows obvious superiority to alluaudite-type $\text{Na}_{2+2x}\text{Fe}_{2-x}(\text{SO}_4)_3$ (Supplementary table 6). For instance, Chou’s group reported that the alluaudite-type $\text{Na}_2\text{Fe}_2(\text{SO}_4)_3$ materials without visible $\text{Na}_6\text{Fe}(\text{SO}_4)_3$ impurity featured a diffusion coefficients range from 10^{-12} to $10^{-10.8}$ $\text{cm}^2 \text{s}^{-1}$ (*Adv. Energy Mater.*, **2018**, 8, 1800944).

Thus, a superior Na^+ conductivity can be determined in $\text{Na}_6\text{Fe}(\text{SO}_4)_4$ phase, which supports our theoretical calculation and electrochemical results.

Figure R9. The characterizations of as-prepared $\text{Na}_6\text{Fe}(\text{SO}_4)_4$ material. (a) XRD pattern. (b) EIS plot of prepared $\text{Na}_6\text{Fe}(\text{SO}_4)_4$ electrode after initial cycle (line: fitted data; dot: pristine data). (c) Fitted $Z' - \omega^{-1/2}$ curve. (d) GITT curves. (e) Calculated Na^+ diffusion coefficients.

On page 5 of revised manuscript, we added the discussion that “Furthermore, the diffusion coefficients range of as-prepared $\text{Na}_6\text{Fe}(\text{SO}_4)_4$ phase is determined by $10^{-10.543}$ to $10^{-7.213}$ $\text{cm}^2 \text{s}^{-1}$ (Supplementary Fig. 9), which shows obvious superiority to pure alluaudite-type $\text{Na}_{2+2x}\text{Fe}_{2-x}(\text{SO}_4)_3$.^{[25]”}

On page 12 of revised manuscript, we added the preparation process of $\text{Na}_6\text{Fe}(\text{SO}_4)_4$ material as “For the preparation of $\text{Na}_6\text{Fe}(\text{SO}_4)_4$ material, the molar ratio (3:1) of anhydrous Na_2SO_4 , FeSO_4 and 10 wt.% carbon black materials (Super P Li) are put into the tank in an environment filled with pure Ar gas (99.99%), and then ball milling is carried out. This process lasts for 6 h, and the speed of ball mill is set to 500 rpm/min. The obtained material is then calcined in a tubular furnace at 350 °C for 12 h with a heating rate of 1 °C/min in Ar gas. Finally, the $\text{Na}_6\text{Fe}(\text{SO}_4)_4$ material is obtained.”

On page 10 of revised Supplementary information, we added the Figure R9 in **Supplementary Fig. 9**.

On page 31 of revised Supplementary information, we added the discussion as **Supplementary Note 2**.

Q2. Lines 138-145. Energy barriers for $\text{Na}_{2.26}\text{Fe}_{1.87}(\text{SO}_4)_3$ phase (3.33 eV) seem to be overestimated and differ significantly from previously published values (0.63 eV) by L.L. Wong *Phys.Chem.Chem.Phys.* 2015, 17, 9186. Computer model should be corrected, differences should be acknowledged and adequately explained.

Our response: We appreciate the reviewers for the significant comments. Accordingly, we have studied the literature carefully and the reasons for the differences on energy barriers of Na⁺ migration inside the crystals, which may be attributed to the calculation software and the nature of crystal structures.

(1) Calculation software. In the literature (*Phys.Chem.Chem.Phys.*, **2015**, 17, 9186), the authors used **softBV software** to calculate the ion migration energy within the Na_{2.25}Fe_{1.875}(SO₄)₃ crystals. While in this work, related calculations are performed in the geometry-based ion-transport analysis library (CAVD), which was carried out on **Computing and Date Platform for Electrochemical Energy Storage Materials**. In order to verify the effect of calculation platform on Na-ion diffusion, we tried softBV software to calculate the ion migration barrier of Na_{2.26}Fe_{1.87}(SO₄)₃ and Na₆Fe(SO₄)₄ crystals in this work (**Figure R10**). Migration pathways are analyzed as regions of low E_{BVSE(Na)} in grids spanning the structure model with a resolution of ca. 0.1 Å³. Visibly, the Na_{2.26}Fe_{1.87}(SO₄)₃ crystal provides intermittent and narrow channels for Na⁺ migration along the b-axis and c-axis direction across a zigzag path between two equivalent positions, while the Na₆Fe(SO₄)₄ crystal possesses successive and broad Na⁺ migration pathways along the 3D (a, b, c-axis) directions, which is consistent with the result obtained from high-throughput computational platform for battery materials. Further calculations on energy barrier reveal that Na⁺ migrates mainly along 1D and 2D directions for both Na_{2.26}Fe_{1.87}(SO₄)₃ and Na₆Fe(SO₄)₄ crystals. The Na_{2.26}Fe_{1.87}(SO₄)₃ crystal shows the Na⁺ transfer barrier of 0.891 eV (along 1D paths), 1.437 eV (along 2D paths) and 999 eV (along 3D paths), respectively. While the Na₆Fe(SO₄)₄ crystal shows the Na⁺ transfer barrier of 0.542 eV (along 1D paths), 0.751 eV (along 2D paths) and 1.659 eV (along 3D paths), respectively. A lower barrier for Na⁺ migration (0.52 eV for Na₆Fe(SO₄)₄ vs. 1.43 eV for Na_{2.26}Fe_{1.87}(SO₄)₃) within Na1–Na2 pathways also supports the superionic conductor property of Na₆Fe(SO₄)₄. Differently, compared with the result obtained from high-throughput computational platform for battery materials, **the calculated values of Na⁺ transfer barrier via SoftBV software are lower**, which is closer to the values (1.38 eV within Na1–Na2 pathway in Na_{2.25}Fe_{1.875}(SO₄)₃ reported in the literature (*Phys.Chem.Chem.Phys.*, **2015**, 17, 9186). Therefore, it is reasonably concluded that the calculation platform should take major responsibility for the difference on energy barriers of Na⁺ migration between this work and reported literature.

The reasons for the calculation differences on various software platforms may be attributed to calculation details or parameters. Although two software platforms are functioned based on the bond valence site energy (BVSE) method, there are still differences in some calculation parameters. For example, in the calculation of energy landscape, the Computing and Date Platform uses a universally fixed scaling factor 0.74 for the radii sum of the interacting ions in the screening factor ρ_{M1-M2} , whereas the softBV software iteratively adapts the screening factor based on the balance between Morse and Coulomb interactions in the individual structure (*Sci. data*, **2020**, 7, 151). Another difference is that softBV analyses migration barriers between local minima of the energy landscape irrespective of their site occupancy leading to a focus on comprehensively mapping interstitial sites, while the BVSE approach performed in the Computing and Date Platform is primarily meant to guide the first principles calculation of energy barriers between the occupied sites in the crystal structure reducing the need to explicitly classify and analyze interstitial sites. Despite these differences, the results calculated by two software confirmed better ionic conductivity of Na₆Fe(SO₄)₄ phase.

(2) Crystal structures. Except calculation software, the crystal structures between this work and report literatures might cause some difference on energy barriers of Na⁺ migration. Accordingly, we carried out the same calculation on the crystal structures in reported literatures on Computing and Date Platform for Electrochemical Energy Storage Materials. In the literature (*Phys.Chem.Chem.Phys.*, **2015**, 17,

9186), the authors used the crystal data from the literature (*Nat. Commun.*, **2014**, 5, 4358). It provides two possible crystal structures (named as **ref.1** and **ref.2**), in which the **ref. 1** is used in calculating the Na-ion diffusion, and the **ref. 2** is the another guessed crystal structure. As shown in **Figure R11** and **Table R1**, three crystals display similar chemical formulas. However, their crystal structures are different, especially for $\text{Na}_{2.26}\text{Fe}_{1.87}(\text{SO}_4)_3$ in this work and $\text{Na}_{2.263}\text{Fe}_{1.868}(\text{SO}_4)_3$ (Ref. 1). The calculation result is presented in **Figure R12**. Taking the 1D migration pathway as an example, the Na3–Na3 transfer displays a higher energy barrier of 3.39 eV for $\text{Na}_{2.26}\text{Fe}_{1.87}(\text{SO}_4)_3$ crystals in this work, compared with those of 0.72 eV for $\text{Na}_{2.263}\text{Fe}_{1.868}(\text{SO}_4)_3$ crystals (ref. 1) and 0.87 eV for $\text{Na}_{2.256}\text{Fe}_{1.872}(\text{SO}_4)_3$ crystals (ref. 2). Therefore, it can be reasonable to infer that, although similar chemical formulas, **the intrinsic difference on crystal structures** cause the partial difference on the migration barriers for Na-ion diffusion within their crystals.

In a word, we think the reasons for the differences on energy barriers of Na^+ migration inside the crystals should be reasonably attributed to the calculation software and crystal structures.

Figure R10. The Na^+ migration channels (blue-green isosurface) revealed by BVSE calculation, Na^+ migration barriers along different directions and energy profiles of path Na1–Na2 for (a, b, c) $\text{Na}_{2.26}\text{Fe}_{1.87}(\text{SO}_4)_3$ and (d, e, f) $\text{Na}_6\text{Fe}(\text{SO}_4)_4$ crystals, respectively. The BVSE calculation was performed by softBV software, and the resolution is 0.1.

Figure R11. The crystal structure of three sulphate cathodes and their crystal parameters.

Figure R12. The calculated Na^+ migration barrier along Na3–Na3 paths within three crystals, respectively. The calculation was performed on the geometry-based ion-transport analysis library (CAVD), which was carried out on Computing and Data Platform for Electrochemical Energy Storage Materials.

Table R1. The atom positions within three crystals.

Atoms	This work			Ref. 1			Ref. 2		
	x	y	z	x	y	z	x	y	z
Na1	0.5000	0.7266	0.7500	0.2435	0.5156	0.5065	0.5000	0.7333	0.7500
Na2	0.0000	0.0000	0.0000	0.7436	0.2424	0.2660	0.0000	0.0000	0.0000
Na3	0.5000	0.9570	0.2500	0.2393	0.2365	0.4650	0.5000	0.9847	0.2500
Fe1	0.7232	0.1572	0.1362	0.4832	0.0897	0.3356	0.7309	0.1579	0.1472
Fe2	/	/	/	0.0215	0.4052	0.1671	/	/	/
S1	0.0000	0.7821	0.1362	0.0129	0.1433	0.1434	0.0000	0.7764	0.7500
S2	0.7469	0.5909	0.8710	0.4884	0.3498	0.3583	/	/	/
S3	/	/	/	0.2477	0.5272	0.9942	/	/	/
O1	0.0970	0.8310	0.7030	0.8955	0.1728	0.0655	0.0844	0.8456	0.7181
O2	0.4560	0.2090	0.5530	0.0872	0.0927	0.9989	0.4457	0.2094	0.5469
O3	0.7471	0.6510	0.6770	0.0167	0.074	0.3352	0.7655	0.6692	0.6841
O4	0.3545	0.0014	0.3760	0.0616	0.2519	0.1892	0.3198	0.9958	0.3765
O5	0.4020	0.5708	0.6760	0.4849	0.4149	0.1684	0.3601	0.5869	0.6708

O6	0.3170	0.1468	0.0370	0.4226	0.2569	0.2992	0.3252	0.1578	0.0837
O7	/	/	/	0.4367	0.4089	0.5165	/	/	/
O8	/	/	/	0.6144	0.3493	0.4441	/	/	/
O9	/	/	/	0.1612	0.6014	0.8906	/	/	/
O10	/	/	/	0.2018	0.4604	0.1581	/	/	/
O11	/	/	/	0.3119	0.4570	0.8609	/	/	/
O12	/	/	/	0.3287	0.5900	0.1259	/	/	/

On page 5 of revised manuscript, we have added the discussion with “The calculation result from another calculation software (softBV) displays similar trend but lower barrier values (Supplementary Fig. S8).”

On page 14 of revised manuscript, we have added the computational details in Theoretical calculation methods with “For comparison, the migration pathways of mobile Na⁺ were also calculated using SoftBV software with a resolution of ca. 0.1 Å³.^{[50, 51]”}

On page 16 of revised manuscript, we have added the references with “[50] H. Chen, L. L. Wong, S. Adams, *Acta Crystallogr. Sect. B Struct. Sci. Cryst. Eng. Mater.* **2019**, 75, 18. [51] L. L. Wong, K. C. Phuah, R. Dai, H. Chen, W. S. Chew, S. Adams, *Chem. Mater.* **2021**, 33, 625.”

On page 9 of revised Supplementary information, we added the Figure R10 as **Supplementary Fig. 8**.

On page 30 of with revised Supplementary information, we added related discussion as **Supplementary Note 1**.

Q3. Lines 177-183, Figure 2f. Position of the diffraction peaks related to the Na₆Fe(SO₄)₄ phase marked with a dashed line between 26 and 27 degrees are highly speculative, and as such this part should be omitted from the article. Recent work (J. Mater. Chem. A, 2019, 7, 13197) revealed very low capacity for the Na₆Fe(SO₄)₄ phase <15 mAh/g.

Our response: Many thanks for your kind consideration. Indeed, previous spectra show poor quality in Figure 2f, which were extracted from the *in-situ* XRD patterns at various voltages. Due to the shielding effect of Be window, the peak signals around 26° show poor quality and strong background noise, which might provide inaccurate information of peak evolution. Accordingly, we further supplemented relevant *ex-situ* XRD patterns of NFS-H at various voltages in **Figure R13** for better credibility of the data and discussion in the revised manuscript.

At pristine, the peak at 25.87° is attributed to the (21-2) plane of Na₆Fe(SO₄)₄ phase (PDF 29-1218). During charged process of Na₆Fe(SO₄)₄, a visible shift to 26.04° around 3.5 V to high degree reveals the emergence of the desodiated Na_{6-x}Fe(SO₄)₄ phase. During subsequent electrochemical process, it kept a gentle trend of peak evolution and finally returned to the initial peak position at 25.89° upon the completion of fully sodiated Na_{2.26-x}Fe_{1.87}(SO₄)₃. For Na_{2.26}Fe_{1.87}(SO₄)₃ phase, its peak attributed to the (13-1) plane shows a weak shift to high angles during the whole charge process, and then backs to the original positions upon the completion the full sodiation of Na_{2.26-x}Fe_{1.87}(SO₄)₃. The bend of peak evolution is later than that of Na₆Fe(SO₄)₄ phase, around 4.2 V. Importantly, at high voltages, the peak intensity becomes significantly weaker, which may be attributed to the severe distortion in the crystal structure due to continuous deintercalation of Na⁺ ions. Similar phenomenon is also reported in Na₂Fe₂(SO₄)₃@C@GO (*Adv. Energy Mater.*, **2018**, 8, 1800944) and Na_{2.4}Fe_{1.8}(SO₄)₃ (*J. Energy Chem.*, **2021**, 54, 564 – 570).

As indicated by reviewer, the Na₆Fe(SO₄)₄ phase has a low sodium storage capacity, which is also

supported by our experiments in **Figure R1** and reported literature (*J. Mater. Chem. A*, **2019**, 7, 13197). Nevertheless, combined with the HAADF-STEM images of NFS-H at various states (**Figure 2** in the revised manuscript) and the electrochemical characteristics (the gauffer in charge-discharge curves at high currents and the 3.05/2.89 V pair in CV curves, **Figure R7**), we believe the $\text{Na}_6\text{Fe}(\text{SO}_4)_4$ phase in NFS-H has a phase transformation between $\text{Na}_6\text{Fe}(\text{SO}_4)_4$ phase and its desodiated $\text{Na}_{6-x}\text{Fe}(\text{SO}_4)_4$ phase. We would therefore like to retain this revised content. We think that the phase transition mechanism is extremely important to improve the understanding on function mechanism of two phases (high-voltage $\text{Na}_{2.26}\text{Fe}_{1.87}(\text{SO}_4)_3$ and ionic-conductive $\text{Na}_6\text{Fe}(\text{SO}_4)_4$ phase) in NFS-H.

Figure R13. Structure evolution of NFS-H. (a) Charge-discharge curve. (b) Newly supplemental *ex-situ* XRD patterns at various voltages. (c) Previous selected *in-situ* XRD patterns at various voltages. (d) *In-situ* XRD patterns. The asterisk represents the $\text{Na}_6\text{Fe}(\text{SO}_4)_4$ material.

On page 6 of revised manuscript, we have revised the discussion with “*Ex-situ* and *in-situ* XRD patterns further demonstrate their structure evolution (Fig. 2g and 2h). During charged process of $\text{Na}_6\text{Fe}(\text{SO}_4)_4$, a subtle shift to high degree marks the emergence of the desodiated $\text{Na}_{6-x}\text{Fe}(\text{SO}_4)_4$ phase, which continues the shift trend until 3.5 V. After that, characteristic reflections of desodiated $\text{Na}_{2.26-x}\text{Fe}_{1.87}(\text{SO}_4)_3$ appear with the canting of lattice planes, including (13-1), (11-2), (400) and (041) planes. Interestingly, with the beginning of the $\text{Na}_{2.26-x}\text{Fe}_{1.87}(\text{SO}_4)_3$ desodiation, the diffraction peak of $\text{Na}_6\text{Fe}(\text{SO}_4)_4$ shifts toward low degree during charged process, indicating partial Na^+ sodiation from $\text{Na}_{2.26-x}\text{Fe}_{1.87}(\text{SO}_4)_3$ to $\text{Na}_6\text{Fe}(\text{SO}_4)_4$. During subsequent electrochemical process, it kept a gentle trend of peak evolution and finally returned to the initial peak position upon the completion of fully sodiated $\text{Na}_{2.26-x}\text{Fe}_{1.87}(\text{SO}_4)_3$.”.

On page 13 of revised manuscript, we have added the experimental detail about *ex-situ* XRD patterns in Materials characterization with “The working electrode was fabricated by mixing the active material, super P and polyvinylidene fluoride (PVDF) with a weight ratio of 80:10:10 in N-methyl-2-pyrrolidone (NMP) to form a slurry. The obtained slurry was coated on ultra-thin aluminum foil (6 μm , Qiandingli electronic technology company) and dried overnight at 120 $^{\circ}\text{C}$ in vacuum with a loading about 6.0–7.0 mg cm^{-2} . Other assembled parameters are the same as those of coin cells. *Ex-situ* XRD pattern was performed in X-ray powder diffraction (XRD, BRUKER D8 ADVANCE A25) with Cu K α radiation ($\lambda=1.54056 \text{ \AA}$) at a scan rate of 1 $^{\circ}/\text{min}$. The electrode was prepared by the same process as above, and was washed by propylene carbonate solvent. The processes of battery disassembly, electrode’s washing and drying are carried out in an argon-filled glove box (O_2 and $\text{H}_2\text{O} < 0.1 \text{ ppm}$).”.

On page 19 of revised manuscript, we have replaced the *ex-situ* XRD patterns with newly supplemental *ex-situ* XRD patterns at various voltages in **Fig. 2**.

REVIEWERS' COMMENTS

Reviewer #1 (Remarks to the Author):

In view of the extensive revision and additional experiments carried out by the authors, the revised manuscript has improved significantly to merit publication in the journal Nature Communications.

Reviewer #2 (Remarks to the Author):

After reading the authors' letter with their response, as well as the corrected manuscript I recommend to accept the manuscript for publication in Nature Communications in its current form.